

# Total energy and potential enstrophy conserving schemes for the shallow water equations using Hamiltonian methods: Derivation and Properties (Part 1)

Christopher Eldred[1] and David Randall[2]

[1]LAGA, University of Paris 13
[2]Department of Atmospheric Science, Colorado State University

*Correspondence to:* Christopher Eldred (chris.eldred@gmail.com)

**Abstract.** The shallow water equations provide a useful analogue of the fully compressible Euler equations since they have similar characteristics: conservation laws, inertia-gravity and Rossby waves and a (quasi-) balanced state. In order to obtain realistic simulation results, it is desirable that numerical models have discrete analogues of these properties. Two prototypical examples of

such schemes are the 1981 Arakawa and Lamb (AL81) C-grid total energy and potential enstrophy conserving scheme, and the 2007 Salmon (S07) Z-grid total energy and potential enstrophy conserving scheme. Unfortunately, the AL81 scheme is restricted to logically square, orthogonal grids; and the S07 scheme is restricted to uniform square grids. The current work extends the AL81 scheme to arbitrary non-orthogonal polygonal grids and the S07 scheme to arbitrary orthogonal spherical

polygonal grids in a manner that allows both total energy and potential enstrophy conservation, by combining Hamiltonian methods (work done by Salmon, Gassmann, Dubos and others) and Discrete Exterior Calculus (Thuburn, Cotter, Dubos, Ringler, Skamarock, Klemp and others). Detailed results of the schemes applied to standard test cases are deferred to Part 2 of this series of papers.

## 1 Introduction

Consider the motion of a (multi-component) fluid on a rotating spheroid under influence of gravity and radiation. This is the fundamental subject of inquiry for geophysical fluid dynamics, covering fields such as weather prediction, climate dynamics and planetary atmospheres. Central to our current understanding of these subjects is the use of numerical models to solve the otherwise intractable equations (such as the fully compressible Euler equations) that result. As a first step towards devel-

oping a numerical model for simulating geophysical fluid dynamics, schemes are usually developed





for the rotating shallow water equations (RSWs). The RSWs provide a useful analogue of the fully compressible Euler equations since they have similar conservation laws, many of the same types of waves and a similar (quasi-) balanced state. It is desirable that a numerical model posses as least some these same properties (see Figure 1, and the discussion in Staniforth and Thuburn (2012)).

In fact, there exists some evidence (Dubinkina and Frank (2007)) that schemes without the appropriate conservation properties can fail to correctly capture long-term statistical behaviour, at least for simplified models without any dissipative effects. However, questions remain as to the relative importance of various conservation properties for a full atmospheric model, especially in the presence of forcing and dissipation (Thuburn (2008)). This subject deserves further study, but a key first

step is the development of a numerical scheme that posses the relevant conserved quantities; and is capable of being run at realistic resolutions on the types of grids that are used in operational weather and climate models.

A pioneering scheme developed over 30 years ago possesses many of these properties (including both total energy and potential enstrophy conservation): the 1981 Arakawa and Lamb scheme

(AL81, Arakawa and Lamb (1981)). Unfortunately, this scheme is restricted to logically square, orthogonal grids such as the lat-lon or conformal cubed-sphere grid. These grids are not quasi-uniform under refinement of resolution, and this leads to clustering at typically target resolutions for next generation weather and climate models (such as 2-3km for weather; and 10-15km for climate). Such clustering will introduce strong CFL limits, and in the case of the lat-lon grid requires polar filtering

(which is not scalable on current computational architectures) in order to take realistic time steps. For these reasons, it is desirable to be able to use quasi-uniform grids such as the icosahedral grid (orthogonal but non-square) or gnomic cubed-sphere (square but non-orthogonal). In addition to the restriction to logically square, orthogonal grids, the AL81 scheme also suffers from poor wave dispersion properties when the Rossby radius is underresolved (Randall (1994)). In fact, the unavoid-

able averaging required for the Coriolis term in a C grid scheme is expected to lead to poor wave dispersion properties for an underresolved Rossby radius regardless of the specific discretization employed.

Recently, there has been an effort to extend the AL81 scheme to more general grids, using tools from discrete exterior calculus (commonly referred to as the TRiSK scheme, Thuburn et al. (2009),

Ringler et al. (2010), Thuburn and Cotter (2012), Weller (2013), Thuburn et al. (2013)). This has lead to the development of a family of schemes on general non-orthogonal (spherical) polygonal meshes that posses all of the desirable properties of AL81 except for: extra modes branches on non quadrilateral meshes, which are unavoidable for C grid schemes; and lack of either total energy or potential enstrophy conservation. It is possible to obtain one or the other, but not both at the

same time. Along different lines, Salmon (Salmon (2004)) showed that AL81 and other doubly-conservative schemes (such as Takano and Wurtele (1982)) are all members of a another family





of schemes on logically square orthogonal meshes. This was done using tools from Hamiltonian methods, which are an area of active research in atmospheric model development.

As an alternative to the AL81 scheme that preserves many of its valuable mimetic properties,
but has good wave dispersion properties independent of Rossby radius, Randall (1994) introduced a scheme for uniform square grids based on the vorticity-divergence formulation (termed the Z grid) of the continuous equations. Subsequently, this approach was extended to arbitrary (spherical) orthogonal polygonal grids with a triangular dual in Heikes and Randall (1995a) and Heikes and Randall (1995b), which included the important case of an icosahedral-hexagonal grid. Although this scheme
posses many of the desirable properties from AL81, it does not conserve total energy or potential enstrophy. However, a similar Z grid scheme based on a Helmholtz decomposition of the momentum instead of the wind that does conserve both total energy and potential enstrophy was developed by Salmon (Salmon (2007),Salmon (2005)) using techniques from Hamiltonian mechanics (specifically, Nambu brackets). The idea of using Hamiltonian mechanics to derive conservative models
for atmospheric dynamical cores has seen a great deal of interest and progress in the past 10 years (see (Gassmann and Herzog (2008),Gassmann (2013),Sommer and Névir (2009),Nevir and Sommer (2009)Dubos and Tort (2014),Dubos et al. (2015),Tort et al. (2015),Salmon (1988),Shepherd (2003)).). With the recent development of Hamiltonian formulations for essentially all of the equation sets and vertical coordinates used in atmospheric dynamics, it seems likely that this approach
will continue to be employed in the future. Unfortunately, the scheme in S07 is defined only for planar grids, and in the key case of general polygonal grids no expression for discrete Hamiltonian or Casimirs was given. This precludes its further development for implementation into an operational dynamical core.

This work combines the discrete exterior calculus approach from Thuburn and Cotter (2012) and
the Hamiltonian approach from Salmon (2004) to extend AL81 to general non-orthogonal (spherical) polygonal grids in a manner that conserves both total energy and potential enstrophy; and to extend S07 to arbitrary (spherical) orthogonal polygonal grids. The extension of AL81 is done through the development of a new $\mathbf{Q}$ (the discretization of $q\hat{k}\times$, which is also known as the nonlinear potential vorticity flux) operator, using tools from Hamiltonian methods. S07 is extended by combining
the Nambu bracket based approach from Salmon (2007) with the discrete exterior calculus tools introduced in Thuburn and Cotter (2012). It should be noted that this work deals only with spatially conservative discretization. Conservation errors introduced due to time discretization are typically much smaller than those due to space discretization. However, the extension of this approach to fully conservative discretization would be a useful contribution.

The remainder of this paper is structured as follows: Section 2 introduces the rotating shallow water equations in both their familiar vector-invariant form and the less familiar Hamiltonian forms. Section 3 presents a family of C grid numerical schemes that posses many of the desirable properties, and discusses the specific member of this family introduced here. Section 4 introduces the new





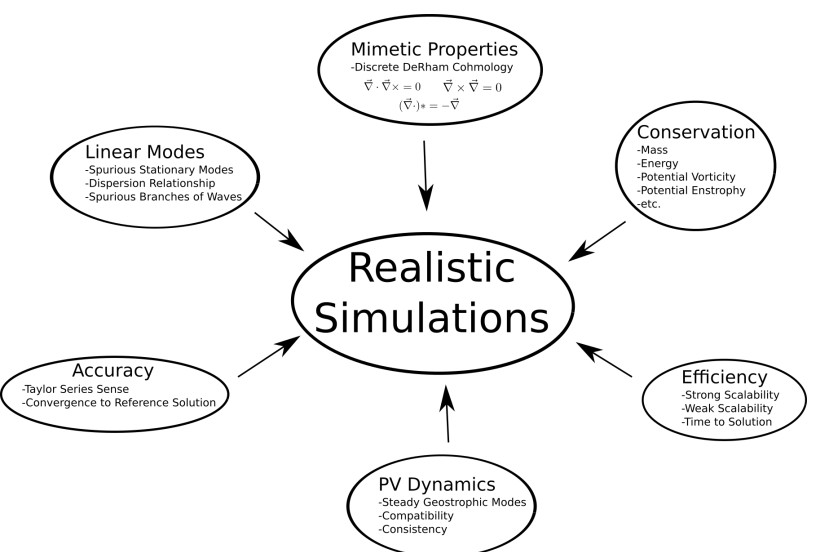

**Figure 1.** A diagram of some desirable model properties for the shallow water equations, organized thematically into groups. Similar considerations apply for the Euler, hydrostatic primitive and other equation sets used in atmospheric models. There is vigorous discussion in the literature and between model designers about the importance of various properties for different applications (such as weather forecasting or long-term climate prediction). The schemes presented here satisfy all of these properties, with the exception of accuracy. There are additional desirable model properties, such as consistent physics-dynamics coupling, compatible and accurate tracer advection, and tractable treatment of acoustic waves that are not presented.

operator **Q** that enables the conservation of both total energy and potential enstrophy in the C grid

scheme. Section 5 presents the Z grid scheme and discusses its key mimetic and conservation properties. Finally, some conclusions (Section 6) are drawn. The appendices discuss various ancillary topics such as the computational grid used (Appendix A), the specific discrete operators employed (Appendices B, C and D), and the discrete variables used in the C and Z grid schemes (Appendices E and F).

## 100  2   Rotating Shallow Water Equations

The rotating shallow water equations (RSWs) for both planar and spherical domains are presented below in several forms: the vector invariant formulation, the vorticity-divergence formulation, the symplectic Hamiltonian formulation based on the vector-invariant form and both Poisson bracket and Nambu bracket formulations based on the vorticity-divergence formulations. Although all of

these formulations are equivalent in the continuous case, they lead to very different discretizations.





### 2.1 Vector Invariant Formulation

The mass continuity equation for the RSWs is expressed in vector invariant form as:

$$\frac{\partial h}{\partial t} + \boldsymbol{\nabla} \cdot (\boldsymbol{F}) = 0 \tag{1}$$

where $h$ is the fluid height and $\boldsymbol{u}$ is the fluid velocity. Similarly, the momentum equation is expressed as:

$$\frac{\partial \boldsymbol{u}}{\partial t} + q\hat{k} \times (\boldsymbol{F}) + \boldsymbol{\nabla}\Phi = 0 \tag{2}$$

where $\boldsymbol{F} = h\boldsymbol{u}$ is the mass flux, $q = \frac{\eta}{h}$ is the potential vorticity, $\eta = \zeta + f$ is the absolute vorticity, $\zeta = \hat{k} \cdot \boldsymbol{\nabla} \times \boldsymbol{u}$ is the relative vorticity, $f$ is the Coriolis force, $\Phi = gh + K + gh_s$ is the Bernoulli function, $h_s$ is the topography height, $g$ is gravity and $K = \frac{\boldsymbol{u} \cdot \boldsymbol{u}}{2}$ is the kinetic energy.

### 2.2 Poisson Bracket Formulation (Vector Invariant)

As discussed in Salmon (2004), let the Hamiltonian $\mathcal{H}$ be given by

$$\mathcal{H} = \int_\Omega \frac{1}{2}\left(h|\boldsymbol{u}|^2\right) + \frac{1}{2}gh(h + 2h_s)d\Omega \tag{3}$$

and $\boldsymbol{x} = (h, \boldsymbol{u})$. Then the time evolution of an arbitrary functional $\mathcal{F}$ can be written as

$$\frac{d\mathcal{F}}{dt} = \{\mathcal{F}, \mathcal{H}\} \tag{4}$$

where the Poisson bracket $\{\mathcal{F}, \mathcal{H}\}$ (which is a bilinear, antisymmetric operator that satifies the Jacobi identity) is

$$\{\mathcal{F}, \mathcal{H}\} = \int_\Omega d\Omega \left(\frac{\delta\mathcal{H}}{\delta\boldsymbol{u}} \cdot \boldsymbol{\nabla}\frac{\delta\mathcal{F}}{\delta h} - \frac{\delta\mathcal{F}}{\delta\boldsymbol{u}} \cdot \boldsymbol{\nabla}\frac{\delta\mathcal{H}}{\delta h} + q\hat{k} \cdot \left(\frac{\delta\mathcal{H}}{\delta\boldsymbol{u}} \times \frac{\delta\mathcal{F}}{\delta\boldsymbol{u}}\right)\right) \tag{5}$$

It is useful to split this into two separate brackets as

$$\{\mathcal{F}, \mathcal{H}\} = \{\mathcal{F}, \mathcal{H}\}_R + \{\mathcal{F}, \mathcal{H}\}_Q \tag{6}$$

where

$$\{\mathcal{F}, \mathcal{H}\}_R = \int_\Omega d\Omega \left(\frac{\delta\mathcal{H}}{\delta\boldsymbol{u}} \cdot \boldsymbol{\nabla}\frac{\delta\mathcal{F}}{\delta h} - \frac{\delta\mathcal{F}}{\delta\boldsymbol{u}} \cdot \boldsymbol{\nabla}\frac{\delta\mathcal{H}}{\delta h}\right) = \int_\Omega d\Omega \left(\frac{\delta\mathcal{H}}{\delta h}(\boldsymbol{\nabla} \cdot \frac{\delta\mathcal{F}}{\delta\boldsymbol{u}}) - \frac{\delta\mathcal{F}}{\delta h}(\boldsymbol{\nabla} \cdot \frac{\delta\mathcal{H}}{\delta\boldsymbol{u}})\right) \tag{7}$$

encompasses the gradient and divergence terms; and

$$\{\mathcal{F}, \mathcal{H}\}_Q = \int_\Omega d\Omega \left(q\hat{k} \cdot \left(\frac{\delta\mathcal{H}}{\delta\boldsymbol{u}} \times \frac{\delta\mathcal{F}}{\delta\boldsymbol{u}}\right)\right) \tag{8}$$

encompasses the nonlinear PV flux term. The functional derivatives $\frac{\delta\mathcal{H}}{\delta\boldsymbol{x}}$ of the Hamiltonian are given by

$$\frac{\delta\mathcal{H}}{\delta\boldsymbol{x}} = \begin{pmatrix} \Phi \\ \boldsymbol{F} \end{pmatrix} \tag{9}$$





This formulation is useful for development of a scheme that posses discrete conservation properties, as discussed below. A functional derivative of some functional $\mathcal{F}[\boldsymbol{x}]$ is defined as

$$\frac{\delta\mathcal{F}}{\delta\boldsymbol{x}} = \lim_{\epsilon\to0}\frac{\mathcal{F}[\boldsymbol{x}+\epsilon\boldsymbol{\phi}]-\mathcal{F}[\boldsymbol{x}]}{\epsilon} \tag{10}$$

## 2.3 Conserved Quantities

Since the rotating shallow water equations form a (non-canonical) Hamiltonian system, we know from Noether's theorem and other considerations (such as the singular nature of the symplectic operator) that there are at least two categories of conserved quantities: Hamiltonian and Casimirs.

### 2.3.1 Energy (Hamiltonian)

The first is simply the Hamiltonian itself. In this case, the Hamiltonian is the total energy of the system. Conservation of the Hamiltonian arises due to the skew-symmetric nature of the Poisson bracket. In particular, using (4) the evolution of $\mathcal{H}$ is given by

$$\frac{d\mathcal{H}}{dt} = \{\mathcal{H},\mathcal{H}\} = -\{\mathcal{H},\mathcal{H}\} = 0 \tag{11}$$

since $\{,\}$ is skew-symmetric. For the rotating shallow water equations, the Hamiltonian is the total
energy of the system. The elegant derivation of energy conservation and its simplicity (relying ONLY on the skew-symmetry of $\{,\}$) motivates the use of the Hamiltonian formulation for development of numerical schemes that conserve energy.

### 2.3.2 Casimirs

The second category of conserved quantities consists of Casimir invariants. Since the rotating shal-
low water equations are a non-canonical Hamiltonian system, the Poisson bracket $\{,\}$ is singular and thus it possesses Casimir invariants $\mathcal{C}$ that satisfy

$$\{\mathcal{F},\mathcal{C}\} = 0 \tag{12}$$

for any functional $\mathcal{F}$. Note that from above, this implies that

$$\frac{d\mathcal{C}}{dt} = 0 \tag{13}$$

For the rotating shallow water equations, the Casimirs take the form

$$\mathcal{C} = \int_{\Omega} hF(q)d\Omega \tag{14}$$

where $F(q)$ is an arbitrary function of the potential vorticity and

$$\frac{\delta\mathcal{C}}{\delta\boldsymbol{x}} = \begin{pmatrix} F(q)-qF'(q) \\ \boldsymbol{\nabla}^{T}F'(q) \end{pmatrix} \tag{15}$$

Important cases include $F = 1$ (mass conservation), $F = q$ (circulation or mass-weighted potential
vorticity) and $F = \frac{q^2}{2}$ (potential enstrophy).



### 2.4 Vorticity-Divergence Formulation

By taking the divergence ($\boldsymbol{\nabla}\cdot$) and curl ($\boldsymbol{\nabla}^{\perp}\cdot$) of (2), we obtain the vorticity-divergence form of the equations:

$$\frac{\partial \zeta}{\partial t} = -\boldsymbol{\nabla} \cdot (\eta \boldsymbol{u}) = -\boldsymbol{\nabla} \cdot (hq\boldsymbol{u}) \tag{16}$$


$$\frac{\partial \mu}{\partial t} = \boldsymbol{\nabla}^{\perp} \cdot (\eta \boldsymbol{u}) - \nabla^2 \Phi = \boldsymbol{\nabla}^{\perp} \cdot (hq\boldsymbol{u}) - \nabla^2 \Phi \tag{17}$$

where $\mu = \boldsymbol{\nabla} \cdot \boldsymbol{u}$ is the divergence. The mass flux can then be split into rotational and divergent components (ie a Helmholtz decomposition) as:

$$h\boldsymbol{u} = (h\boldsymbol{u})_{div} + (h\boldsymbol{u})_{rot} = \boldsymbol{\nabla}\chi + \boldsymbol{\nabla}^{\perp}\psi \tag{18}$$

where $(h\boldsymbol{u})_{div} = \boldsymbol{\nabla}\chi$ and $(h\boldsymbol{u})_{rot} = \boldsymbol{\nabla}^{\perp}\psi$. The streamfunction $\psi$ and velocity potential $\chi$ can be related to the vorticity and divergence as

$$\zeta = \eta - f = \boldsymbol{\nabla} \cdot (h^{-1}\boldsymbol{\nabla}\psi) + J(h^{-1},\chi) \tag{19}$$

$$\mu = \boldsymbol{\nabla} \cdot (h^{-1}\boldsymbol{\nabla}\chi) + J(\psi,h^{-1}) \tag{20}$$

where $J(a,b) = \boldsymbol{\nabla} \cdot (a\boldsymbol{\nabla}^T b) = \boldsymbol{\nabla}^T \cdot (a\boldsymbol{\nabla} b)$ is the Jacobian operator. The Hemholtz decomposition connects the vorticity-divergence formulation and the vector invariant formulations. In the preceding, we have neglected the possibility of a harmonic component (a component $A$ for which $\nabla^2 A = 0$), which works because the harmonic component on the sphere is zero. On the doubly periodic plane, it would be possible to have a constant harmonic component. Finally, (1) and (2) can be re-written

in terms of $\chi$ and $\psi$ directly as

$$\frac{\partial h}{\partial t} = -\nabla^2 \chi \tag{21}$$

$$\frac{\partial \zeta}{\partial t} = J(q,\psi) - \boldsymbol{\nabla} \cdot (q\boldsymbol{\nabla}\chi) \tag{22}$$

$$\frac{\partial \mu}{\partial t} = J(q,\chi) + \boldsymbol{\nabla} \cdot (q\boldsymbol{\nabla}\psi) - \nabla^2 \Phi \tag{23}$$

### 2.5 Poisson Bracket Formulation (Vorticity-Divergence)

As shown in Salmon (2007), the preceding equations (21), (22) and (23) can be also be written in terms of a Poisson bracket. Let $\boldsymbol{x} = (h,\zeta,\mu)$ and define the Hamiltonian

$$\mathcal{H} = \int\limits_{\Omega} \frac{1}{2h}\left(|\boldsymbol{\nabla}\chi|^2 + |\boldsymbol{\nabla}\psi|^2 + 2J(\chi,\psi)\right) + \frac{1}{2}gh(h + 2h_s)d\Omega \tag{24}$$





Note that

$$\delta \mathcal{H} = \int_\Omega d\Omega \left(-\psi \delta \zeta - \chi \delta \mu + \Phi \delta h\right) \tag{25}$$

where

$$\Phi = K + gh = \frac{|\boldsymbol{\nabla}\chi|^2 + |\boldsymbol{\nabla}\psi|^2 + 2J(\chi,\psi)}{2h^2} + gh \tag{26}$$

which gives

$$\frac{\delta \mathcal{H}}{\delta \boldsymbol{x}} = \begin{pmatrix} \Phi \\ -\psi \\ -\chi \end{pmatrix} \tag{27}$$

(this is the functional derivative of the Hamiltonian with respect to $\boldsymbol{x}$). Also define a Poisson bracket (which is bilinear, anti-symmetric and satisfies the Jacobi identity) as

$$\{\mathcal{A},\mathcal{B}\} = \{\mathcal{A},\mathcal{B}\}_{\mu\mu} + \{\mathcal{A},\mathcal{B}\}_{\zeta\zeta} + \{\mathcal{A},\mathcal{B}\}_{\mu\zeta h} \tag{28}$$

where

$$\{\mathcal{A},\mathcal{B}\}_{\zeta\zeta} = \int_\Omega d\Omega q J(\mathcal{A}_\zeta, \mathcal{B}_\zeta) \tag{29}$$

$$\{\mathcal{A},\mathcal{B}\}_{\mu\mu} = \int_\Omega d\Omega q J(\mathcal{A}_\mu, \mathcal{B}_\mu) \tag{30}$$

$$\{\mathcal{A},\mathcal{B}\}_{\zeta\mu h} = \int_\Omega d\Omega q (\boldsymbol{\nabla}\mathcal{A}_\mu \cdot \boldsymbol{\nabla}\mathcal{B}_\zeta - \boldsymbol{\nabla}\mathcal{A}_\zeta \cdot \boldsymbol{\nabla}\mathcal{B}_\mu) + (\boldsymbol{\nabla}\mathcal{A}_\mu \cdot \boldsymbol{\nabla}\mathcal{B}_h - \boldsymbol{\nabla}\mathcal{A}_h \cdot \boldsymbol{\nabla}\mathcal{B}_\mu) \tag{31}$$

for arbitrary functionals $\mathcal{A}$ and $\mathcal{B}$. As before, the time evolution of an arbitrary functional $\mathcal{A}$ is then given by

$$\frac{d\mathcal{A}}{dt} = \{\mathcal{A},\mathcal{H}\} \tag{32}$$

It is easy to see that (21), (22) and (23) are recovered when $\mathcal{A}$ is set equal to $h, \zeta$ or $\mu$, respectively. Note that each of the brackets (29), (30) and (31) are anti-symmetric, and that the Casimirs $\mathcal{C} =$

$\int_\Omega h F(q) d\Omega$ satisfy $\{\mathcal{A},\mathcal{C}\} = 0$ (where $F$ is an arbitrary function and $\mathcal{A}$ is an arbitrary functional) independently for each bracket.

     The use of the Poisson (and Nambu) bracket formulation of the shallow water equations is motivated by the intimate connection between these formulations and the conserved quantities. As is well-known, the conservation of energy $\mathcal{H}$ rests solely on the anti-symmetry of the Poisson bracket,

and a numerical scheme that retains this feature will automatically conserve energy. However, potential enstrophy is a Casimir, and therefore developing a numerical scheme using the Poisson formulation that conserves it requires that the discrete potential enstrophy lies in the null space of the resulting discrete bracket. This can be difficult, especially on arbitrary grids, and this motivates the use of a continuous formulation that does not contain a null space, which is discussed below.





### 2.6 Nambu Bracket Formulation (Vorticity-Divergence)

Fortunately, there is a closely related formulation of the shallow water equations in terms of Nambu brackets (see Salmon (2007)):

$$\{\mathcal{F}, \mathcal{H}, \mathcal{Z}\}_{\zeta\zeta\zeta} = \int_{\Omega} d\Omega \mathcal{Z}_{\zeta} J(\mathcal{F}_{\zeta}, \mathcal{H}_{\zeta}) \tag{33}$$

$$\{\mathcal{F}, \mathcal{H}, \mathcal{Z}\}_{\mu\mu\zeta} = \int_{\Omega} d\Omega \mathcal{Z}_{\zeta} J(\mathcal{F}_{\mu}, \mathcal{H}_{\mu}) \tag{34}$$

$$\{\mathcal{F}, \mathcal{H}, \mathcal{Z}\}_{\mu\zeta h} = \int_{\Omega} d\Omega \left( \boldsymbol{\nabla}\mathcal{Z}_h \cdot \boldsymbol{\nabla}\mathcal{F}_{\mu} \cdot \boldsymbol{\nabla}\mathcal{H}_{\zeta} \cdot \frac{1}{\boldsymbol{\nabla}q} - \boldsymbol{\nabla}\mathcal{Z}_h \cdot \boldsymbol{\nabla}\mathcal{F}_{\zeta} \cdot \boldsymbol{\nabla}\mathcal{H}_{\mu} \cdot \frac{1}{\boldsymbol{\nabla}q} \right) + \mathrm{cyc}(\mathcal{F}, \mathcal{H}, \mathcal{Z}) \tag{35}$$

where cyc is a cyclic permutation, $Z = \int_{\Omega} d\Omega h \frac{q^2}{2}$ is the potential enstrophy, and the multipart dot product is simply the product of the individual components, summed over each basis (for example, in 2D doubly periodic flow the first term is $\frac{\partial_x \mathcal{Z}_h \partial_x \mathcal{F}_{\delta} \partial_x \mathcal{H}_{\zeta}}{\partial_x q}$). The time evolution of an arbitrary functional $\mathcal{A}$ is now given by

$$\frac{d\mathcal{A}}{dt} = \{\mathcal{A}, \mathcal{H}, \mathcal{Z}\} = \{\mathcal{A}, \mathcal{H}, \mathcal{Z}\}_{\zeta\zeta\zeta} + \{\mathcal{A}, \mathcal{H}, \mathcal{Z}\}_{\mu\mu\zeta} + \{\mathcal{A}, \mathcal{H}, \mathcal{Z}\}_{\mu\zeta h} \tag{36}$$

These brackets are useful because they are triply anti-symmetric (which ensures the conservation of $\mathcal{H}$ and $\mathcal{Z}$) and non-degenerate (they have no Casimirs). In fact, discrete conservation of both total energy and potential enstrophy requires only the triply anti-symmetric nature is retained. It is also possible to generalize these brackets to ANY Casimir (as shown in Salmon (2005)), but since we are interested mostly in potential enstrophy conservation this is not necessary. These brackets will form the basis of the Z grid discretization method discussed below.

## 3 C Grid Scheme

Following Thuburn and Cotter (2012), the prognostic variables for the C grid scheme are the mass primal 2-form $m_i$ and the wind dual 1-form $u_e$. These are naturally staggered, since primal 2-forms are associated with primal grid cells and dual 1-forms are associated with dual grid edges. Letting $\boldsymbol{x} = (m_i, u_e)$, the vector-invariant Poisson bracket can be discretized in a manner that preserves its anti-symmetric character (which ensures total energy conservation) and a subset of the Casimir invariants (specifically: mass, potential vorticity and potential enstrophy). Combined with a choice for the discrete Hamiltonian, this constitutes a complete discretization for the nonlinear rotating shallow water equations. Ideally, one would use a Nambu bracket formulation of the vector invariant shallow water equations rather than the Poisson bracket formulation in order to avoid the difficulties





associated with developing a discretization that has the correct Casimirs, since in the Nambu bracket
case only anti-symmetry must be enforced. Unfortunately, the only known Nambu bracket for the
vector invariant shallow water equations possesses intractable singularities and is not suitable as the
basis for developing a discretization (Thuburn and Woollings (2005)).

Specifically, the brackets 7 and 8 are discretized using the operators from Appendices C and B as:

$$
\quad \{\mathcal{A},\mathcal{B}\}_R = -\left(\frac{\delta\mathcal{A}}{m_i}, D_2\frac{\delta\mathcal{B}}{u_e}\right)_{\mathbf{I}} + -\left(\frac{\delta\mathcal{A}}{u_e}, \bar{D}_1\frac{\delta\mathcal{B}}{m_i}\right)_{\mathbf{H}} \tag{37}
$$

$$
\{\mathcal{A},\mathcal{B}\}_Q = \left(\frac{\delta\mathcal{A}}{u_e}, \mathbf{Q}\frac{\delta\mathcal{B}}{u_e}\right)_{\mathbf{H}} \tag{38}
$$

where the discrete functionals (such as $\mathcal{A}$) are expressed as inner products using the Hodge stars.
Note that these discrete brackets are only bilinear and anti-symmetric, they do not satisfy the Jacobi
identity. In addition, they posses only a subset of the Casimirs of the continuous brackets. Therefore
they should be properly be termed quasi-Poisson brackets. The brackets given in (37) and (38) are
essentially a generalization of the brackets introduced in S04 from uniform square grids to arbitrary
polygonal grids, using operators from discrete exterior calculus. The discrete function derivative
with respect to a particular discrete form is the corresponding dual form. For example, consider
$\mathcal{F} = (A_i, B_i)_{\mathbf{I}}$, where $A_i$ and $B_i$ are primal 2-forms. Then $\frac{\delta\mathcal{F}}{\delta A_i} = \mathbf{I}B_i$, which is a dual 0-form. The
Hamiltonian $\mathcal{H}$ is discretized as:

$$
\mathcal{H} = \frac{1}{2}(m_i, gm_i)_{\mathbf{I}} + \frac{1}{2}(u_e, C_e)_{\mathbf{H}} + (m_i, gb_i)_{\mathbf{I}} \tag{39}
$$

where $g$ is the acceleration due to gravity, $C_e = m_e u_e$ and $m_e = \phi\mathbf{I}m_i$. Taking functional derivatives
yields

$$
\quad \frac{\delta\mathcal{H}}{\delta\boldsymbol{x}} = \begin{pmatrix} \Phi_i \\ F_e \end{pmatrix} \tag{40}
$$

where $\Phi_i$ is the Bernoulli function dual 0-form and $F_e$ is the mass flux primal 1-form. Computing
actual values yields: $\Phi_i = \mathbf{I}(K_i + gm_i + gb_i)$ with $K_i = \phi^T\frac{u_e^T\mathbf{H}u_e}{2}$, where $b_i$ is the topographic height
primal 2-form and $K_i$ is the kinetic energy primal 2-form; and $F_e = \mathbf{H}C_e$. A detailed description of
these discrete variables and their staggering on the computational grid can be found in Appendix E,
and a diagram of their staggering is in Figure 2. The resulting discrete evolution equations are

$$
\frac{\partial m_i}{\partial t} + D_2 F_e = 0 \tag{41}
$$

$$
\frac{\partial u_e}{\partial t} - \mathbf{Q}(F_e, q_v) + \bar{D}_1\Phi_i = 0 \tag{42}
$$




In fact, by making alternative choices for $F_e$, $\mathbf{Q}$ and $\Phi_i$ (along with the operators discussed below) it is possible to recover a wide range of C grid schemes present in the literature (such as Ringler et al. (2010), Thuburn et al. (2013) and Weller (2013)), see THESIS for more details). The operators $D_2$, $D_1$, $\bar{D}_1$, $\bar{D}_2$, $\mathbf{I}$, $\mathbf{J}$, $\mathbf{R}$, $\mathbf{W}$ and $\mathbf{H}$ are defined in Appendices B and C (and can also be found in a general form in Thuburn and Cotter (2012)). The novelty of the current scheme is a new definition of $\mathbf{Q}$, such that the properties of total energy conservation, potential enstrophy conservation and steady geostrophic modes hold simultaneously. This is the subject of Section 4.

### 3.1 Linearized Scheme

As is well-known, the linearized version of a Hamiltonian system about a steady state can be found by evaluating the brackets at that state and using the quadratic approximation to the associated psuedo-energy as the Hamiltonian (Shepherd (1993)). Following this procedure and letting the Coriolis force $f$ be a constant, $b_i = 0$ and assuming a background state of $\boldsymbol{x} = (H, 0)$, we obtain

$$\{\mathcal{A}, \mathcal{B}\}_R = -\left(\frac{\delta \mathcal{A}}{m_i}, D_2 \frac{\delta \mathcal{B}}{u_e}\right)_{\mathbf{I}} + -\left(\frac{\delta \mathcal{A}}{u_e}, \bar{D}_1 \frac{\delta \mathcal{B}}{m_i}\right)_{\mathbf{H}} \tag{43}$$

$$\{\mathcal{A}, \mathcal{B}\}_W = \frac{f}{H}\left(\frac{\delta \mathcal{A}}{u_e}, \mathbf{W} \frac{\delta \mathcal{B}}{u_e}\right)_{\mathbf{H}} \tag{44}$$

for the brackets (where $\mathbf{W} = \mathbf{Q}_{q_v = 1}$ is the linearized version of $\mathbf{Q}$) and

$$\mathcal{H} = \frac{1}{2}(m_i, g m_i)_{\mathbf{I}} + \frac{1}{2} H (u_e, u_e)_{\mathbf{H}} \tag{45}$$

for the Hamiltonian, which has associated functional derivatives of

$$\frac{\delta \mathcal{H}}{\delta \boldsymbol{x}} = \begin{pmatrix} g \mathbf{I} m_i \\ H \mathbf{H} u_e \end{pmatrix} \tag{46}$$

The resulting evolution equations are

$$\frac{\partial m_i}{\partial t} + H D_2 \mathbf{H} u_e = 0 \tag{47}$$

$$\frac{\partial u_e}{\partial t} - f \mathbf{W} \mathbf{H} u_e + g \bar{D}_1 \mathbf{I} m_i = 0 \tag{48}$$

### 3.2 Properties of Scheme

This scheme has many important properties, including:

1. Mass and potential vorticity conservation: Both mass $m_i$ and mass-weighted potential vorticity $m_v q_v$ are conserved in both a local (flux-form) and global (integral) sense.





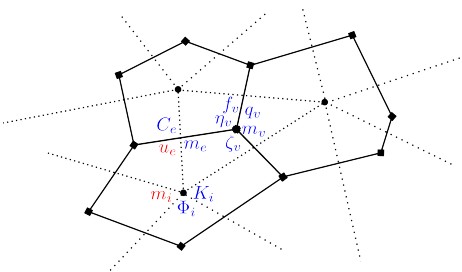

**Figure 2.** A subset of discrete variables and their staggering on the computational grid for the C grid scheme. A subscript $i$ indicates quantities defined at primal grid cells or dual grid vertices, a subscript $e$ indicates quantities defined at primal or dual grid edges, and a subscript $v$ indicates quantities defined at primal grid vertices or dual grid cells. The prognostic (red) quantities are the mass primal 2-form $m_i$ and the wind dual 1-form $u_e$, the other quantities are diagnostic (blue). More details can be found in Appendix E

.

2. No spurious vorticity production: By construction, $D_2 D_1 = 0$ and there is no spurious production of vorticity due to the gradient term in the wind equation.

3. Linear stability (pressure gradient force and Coriolis force conserve energy): This due to the fact that $\mathbf{I}$, $\mathbf{J}$ and $\mathbf{H}$ are all symmetric positive-definite; $D_2^T = -\bar{D}_1$; $\bar{D}_2^T = D_1$ and $\mathbf{W} = -\mathbf{W}^T$.

4. Steady geostrophic modes: By construction, $-\mathbf{R}D_2 = \mathbf{W}\bar{D}_2$ (noting that $\mathbf{W}$ is the same for all members of this family), which gives steady geostrophic modes.

5. PV Compatibility: again by construction $-\mathbf{R}D_2 = \mathbf{W}\bar{D}_2$ with $\mathbf{Q}_{q_v=c} \to c\mathbf{W}$, and therefore the potential vorticity equation is compatible with the diagnostic mass equation (a constant PV field remains constant). Note that this is same as the condition required for steady geostrophic modes.

6. Other conservation properties: see below for a discussion on total energy and potential enstrophy conservation.

Table 1 shows a summary of the required properties in order for the resulting scheme to have all of the mimetic and conservation properties discussed above.

### 3.2.1 Total Energy Conservation

Following S04, total energy will be conserved for any choice of $\mathcal{H}$ if the discrete brackets retain their anti-symmetric character. This requires that $D_2^T = -\bar{D}_1$, and that $\mathbf{Q} = -\mathbf{Q}^T$. The first condition is satisfied by construction of the discrete exterior derivative operators $D_2$ and $\bar{D}_1$. The second condition is satisfied only for certain choices of $\mathbf{Q}$. One example is $\mathbf{Q} = \frac{1}{2} q_e \mathbf{W} + \frac{1}{2} \mathbf{W} q_e$ (as used in





**Table 1.** Summary of required operator properties for obtaining the desirable mimetic properties along with total energy and potential enstrophy conservation. For example $\mathbf{I}$ is a discrete Hodge star that maps from primal 2-forms to dual 0-forms, and must be symmetric positive. The only operator that merits additional explanation here is $\phi$- it is used to construct mass at edges for use in determining the mass flux, and its transpose $\phi^T$ is used for kinetic energy calculations. This ensures that the scheme conserves energy, see Thuburn and Cotter (2012) or THESIS for more details.

| Operator | Properties | Notes | Mapping |
|---|---|---|---|
| $\mathbf{I}$ | Symmetric Positive Definite | Hodge Star | p2 ->d0 |
| $\mathbf{J}$ | Symmetric Positive Definite | Hodge star | d2 ->p0 |
| $\mathbf{H}$ | Symmetric Positive Definite | Hodge star | d1 ->p1 |
| $\mathbf{W}$ | $\mathbf{R}D_2 = \bar{D}_2\mathbf{W}$ | Interior product (contraction) | p1 ->d1 |
| $\mathbf{R}$ | $\mathbf{W} = -\mathbf{W}^T$ | Identity operator | p2 ->d2 |
| $\mathbf{Q}$ | $\mathbf{Q} = -\mathbf{Q}^T$ <br> $\mathbf{Q} \to q_0\mathbf{Q}$ when $q_v = q_0$ is constant <br> $-\bar{D}_1\mathbf{R}^T\frac{q_v^2}{2} + \mathbf{Q}D_1 q_v = 0 \quad \forall q_v$ | Interior product (contraction) | p1 ->d1 |
| $D_2$ | $D_2 D_1 = 0$ and $D_2^T = -\bar{D}_1$ | Exterior Derivative | p1 ->p2 |
| $\bar{D}_2$ | $\bar{D}_2 \bar{D}_1 = 0$ and $\bar{D}_2^T = D_1$ | Exterior Derivative | d1 ->d2 |
| $D_1$ | $D_2 D_1 = 0$ and $D_2^T = -\bar{D}_1$ | Exterior Derivative | p0 ->p1 |
| $\bar{D}_1$ | $\bar{D}_2 \bar{D}_1 = 0$ and $\bar{D}_2^T = D_1$ | Exterior Derivative | d0 ->d1 |
| $\phi$ | see text | see text | see text |

Ringler et al. (2010)), where $q_e$ is any function that, given the set of $q_v$ at primal vertices, computes a unique $q_e$ at primal edges (such as $q_e = \frac{1}{2}\sum_{v \in VE(e)} q_v$). Flexibility in the choice of $q_e$ allows a wide variety of stabilization methods such as CLUST or APVM (Weller (2012) and Weller et al. (2012)). Unfortunately, this choice does not conserve potential enstrophy.

### 3.2.2 Potential Enstrophy Conservation

Following S04, potential enstrophy is a Casimir and therefore will be conserved when

$$\{\mathcal{Z}, \mathcal{A}\} = 0 \qquad (49)$$

holds for any choice of functional $\mathcal{A}$. Note that

$$\mathcal{Z} = (q_v, m_v q_v)_\mathbf{J} = (\eta_v, \eta_v/m_v)_\mathbf{J} \qquad (50)$$

is the potential enstrophy where $q_v$ is potential vorticity primal 0-form, Note that $q_v = \frac{\eta_v}{m_v}$, where $m_v = \mathbf{R}m_i$ is the mass dual 2-form and $\eta_v = \zeta_v + f_v = \bar{D}_2 u_e + f_v$ is the absolute vorticity dual





2-form. Its functional derivatives are

$$\frac{\delta \mathcal{Z}}{\delta \boldsymbol{x}} = \begin{pmatrix} -\mathbf{R}\frac{q_v^2}{2} \\ D_1 q_v \end{pmatrix} \qquad (51)$$

Using the chain rule for functional derivatives, it suffices to show that equation (49) holds for $\mathcal{A} = \sum_i m_i$ and $\mathcal{A} = \sum_e u_e$. Therefore equation (49) reduces to

$$D_2 D_1 q_v = 0 \qquad (52)$$

$$-\bar{D}_1 \mathbf{R}\frac{q_v^2}{2} + \mathbf{Q} D_1 q_v = 0 \qquad (53)$$

which must hold for any choice of $q_v$. The first of these is again satisfied by construction for $D_2$ and $D_1$. The second is much trickier, and is the main subject of section 4. One example is $\mathbf{Q} = q_e \mathbf{W}$ (as used in Ringler et al. (2010)), where $q_e = \frac{1}{2} \sum_{v \in VE(e)} q_v$. Unfortunately, this choice does not conserve total energy. It would be possible to explore alternative definitions of $\mathcal{Z}$, but these would lead to different, less natural stencils for $q_v$.

### 3.3 Arakawa and Lamb 1981

In the case of a uniform square grid, the C scheme grid above reduces to the well-known Arakawa and Lamb 1981 total energy and potential enstrophy scheme (modified to prognose $m_i$ and $u_e$ if their choice of $\mathbf{Q}$ is used. Unfortunately, the definition of $\mathbf{Q}$ presented in AL81 works only for logically square, orthogonal grids. For more general, non-orthogonal polygonal grids, a new operator $\mathbf{Q}$ must be found. This is the subject of the next section.

### 3.4 Hollingsworth Instability

Since this is an extension of Arakawa and Lamb 1981 scheme, it seems extremely likely that the proposed scheme will suffer from the Hollingsworth instability, especially if applied in a height coordinate framework using a Lorenz staggering in the vertical (as discussed in Bell et al. (2016) and Hollingsworth et al. (1983)). It also seems likely that proposed scheme will avoid the Hollingsworth instability when used with an isentropic or Lagrangian vertical coordinate, or when a Charney-Phillips staggering is used in the vertical. If the instability is encountered, it would be simple to modify the stencil of the kinetic energy in a consistent manner (to preserve total energy conservation, by simply modifying the Hamiltonian itself), which has been shown to be sufficient to prevent the instability (Hollingsworth et al. (1983)). Therefore, the possible presence of the instability is not expected to prevent use of this scheme in a full 3D model.





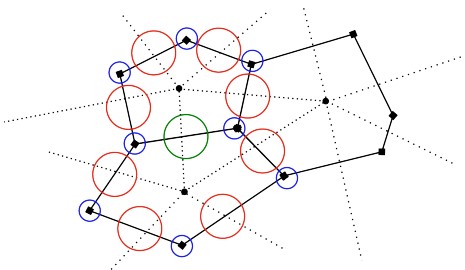

**Figure 3.** A diagram of the stencil of $\mathbf{Q}$ when applied to an edge $e$ (green). The nonlinear PV flux $\mathbf{Q}F_e$ at edge $e$ (green) is a linear combination of the mass fluxes $F_e$ at the edges $e' \in ECP(e)$ (red), where the weights $\alpha_{e,e',v}$ are themselves a linear combination of the potential vorticity $q_v$ at vertices $v \in VC(i)$ (blue) and $i$ is the cell shared between edges $e$ (green) and $e'$ (red). By choosing the weights $\alpha_{e,e',v}$ appropriately, an operator $\mathbf{Q}$ can be found that simultaneously conserves both total energy and potential enstrophy; and supports steady geostrophic modes.

## 4    Operator Q

The principal novelty of the new C grid scheme is the specification of a $\mathbf{Q}$ operator that simultaneously conserves total energy and potential enstrophy, and also supports PV compatibility. Previous work found choices for $\mathbf{Q}$ that conserved either total energy or potential enstrophy, but not both. The key lies in S04, showing that the AL81 approach could be extended to more general stencils (although retaining a logically square, orthogonal grid). This work takes the Salmon 2004 approach in a different direction, keeping the same stencil as AL81 but considering a general polygonal grid.

### 4.1    Definition of Q

Loosely following S04, define $\mathbf{Q}$ as

$$\mathbf{Q}F_e = \sum_{e' \in ECP(e)} \sum_{v \in VC(i)} q_v \alpha_{e,e',v} F_e \tag{54}$$

where $i$ is the primal grid cell covered by both $e$ and $e'$. A diagram of this operator is shown in Figure 3. An equivalent alternative form for $\mathbf{Q}$ given in terms of the Poisson bracket that closely mimics the one found in S04 can be found in the appendix. It is easy to see that in the case of a logically square orthogonal grid, this approach reduces to the same stencil considered by AL81. At this point, the coefficients $\alpha_{e,e',v}$ are undetermined.

### 4.2    Linear System for $\alpha$

It remains to determine the coefficients $\alpha_{e,e',v}$ in a manner such that the resulting operator $\mathbf{Q}$ conserves both total energy and potential enstrophy, and satisfies PV consistency.





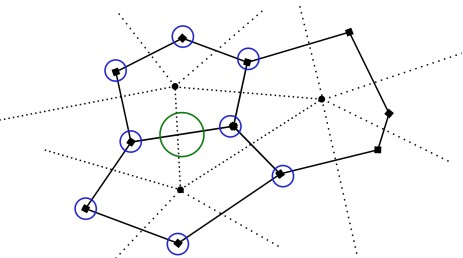

**Figure 4.** A diagram of the stencil $v \in CVE(e) = VE(i1) \cup VE(i2)$ with $(i1, i2) = CE(e)$, which is simply the union of all vertices $v$ (blue) in the cells on either side of edge $e$ (green).

### 4.2.1 Requirements introduced by energy conservation

Following S04, in order for $\mathbf{Q}$ to be energy conserving then $\mathbf{Q} = -\mathbf{Q}^T$. In terms of the coefficients, this implies that $\alpha_{e,e',v} = -\alpha_{e',e,v}$, or in other words, they are anti-symmetric under an interchange of $e$ and $e'$.

### 4.2.2 Requirements introduced by potential enstrophy conservation

   From (53), in order for $\mathbf{Q}$ to conserve potential enstrophy $-\bar{D}_1 \mathbf{R} \frac{q_v^2}{2} + \mathbf{Q} D_1 q_v = 0$ must hold for any
choice of $q_v$. Expanding this out yields

$$\sum_{e' \in ECP(e)} \left( \sum_{v \in EVC(e,e')} \alpha_{e,e',v} q_v \right) \sum_{v' \in VE(e')} t_{e',v'} q'_v = \sum_{i \in CE(e)} (-n_{e,i}) \sum_{v \in VC(i)} R_{i,v} \frac{q_v^2}{2} \tag{55}$$

for every $e$, which must hold for any choice of $q_v$. For a given edge $e$, the vertices in question are $v \in CVE(e)$ (shown in Figure 4) where $CVE(e) = VE(i1) \cup VE(i2)$ and $(i1, i2) = CE(e)$. Both the left and right hand side of these equations are a quadratic form in this set of vertices, and for this
to hold for arbitrary $q_v$ the coefficients in these two quadratic forms must be equal. These coefficients are linear combinations of the $\alpha$'s, and therefore the equality of these quadratic forms implies a set of linear equations for the $\alpha$'s.

   Specifically, for each grid cell $i$ with $n_e$ edges and $n_v$ vertices (note that $n_e = n_v$ for a polygonal grid cell, but it is useful to keep distinct notation to ease exposition), there are $n_e \frac{n_v(n_v+1)}{2}$ equa-
tions (coefficients in the quadratic forms) and $n_v \frac{n_e(n_e-1)}{2}$ unknowns (the coefficients $\alpha_{e,e',v}$). This is therefore an overdetermined system, and the coefficient will be found through a least squares procedure. At least some of the additional freedom will be used to split the equations into independent subset for each grid cell (see below), which makes implementation practical for operational grids. The equations come from equating the coefficients in the two quadratic forms: there are $\frac{n_v(n_v+1)}{2}$
independent vertex pairs, and $n_e$ edges. The unknowns are the coefficients $\alpha_{e,e',v}$ that are associated with the grid cell: there are $\frac{n_e(n_e-1)}{2}$ independent unique edge pairs, and $n_v$ vertices. Note that this





has already taken into account the fact that $\alpha_{e,e',v} = -\alpha_{e',e,v}$ (hence the wording unique edge pair) which reduces the number of independent coefficients in half. Letting $v$ and $v'$ loop over the vertices in the cell (they are the unique members of $VC(i) \times VC(i)$), the equations are given by

$$A_{v,v} = \sum_{e' \in EVE(v,e,i)} \alpha_{e,e',v} t_{e',v} sgn(e,e') \tag{56}$$

$$B_{v,v} = \sum_i n_{e,i} \frac{R_{i,v}}{2} = \frac{R_{i,v}}{2} \tag{57}$$

where the sum for $B_{v,v}$ occurs only when $v \in VE(e)$; and

$$A_{v,v'} = \sum_{e' \in EVE(v',e,i)} \alpha_{e,e',v} t_{e',v'} sgn(e,e') + \sum_{e' \in EVE(v,e,i)} \alpha_{e,e',v'} t_{e',v} sgn(e,e') \tag{58}$$

$$B_{v,v'} = 0 \tag{59}$$

where $e$ loops over each edge in $i$ and $EVE(v,e,i) = EC(i) \cap EV(v) - e$; and $sgn(e,e') = 1 = -sgn(e',e)$ (which ensures that the scheme is also energy conservative). A diagram of $EVE(v,e,i)$ is provided in Figure 5. Note that coefficients in one cell are coupled with adjacent cells when $v \in VE(e)$ or $v' \in VE(e)$; that is to say, the equations involve coefficients that are associated with other grid cells. On a non-uniform mesh, this means that the entire set of coefficients must be solved for at the same time.

The solution procedure outlined above gives a large matrix system

$$\mathbf{A}\boldsymbol{\alpha} = \boldsymbol{b} \tag{60}$$

where each row in $\mathbf{A}$ represents an equation obtained by equating coefficients in the quadratic forms, and $\boldsymbol{\alpha}$ is the vector of unknown coefficients. This system can be solved (via a least-squares approach) to yield a set of coefficients $\boldsymbol{\alpha}$ such that $\mathbf{Q}$ conserves potential enstrophy. This procedure is essentially identical to the one employed in S04; when applied to a uniform square grid it reproduces AL81 and produces a total energy and potential enstrophy conserving scheme on a uniform hexagonal grid (not shown, verified numerically). In addition, the coefficients only have to be computed once, and then stored for later use. Unfortunately, the system that results from this procedure is impractical to solve for realistic non-uniform meshes: it is too large and ill-conditioned. For example, on an icosahedral-hexagonal mesh with O(1 million) grid cells, there will be O(90 million) coupled coefficients that need to be solved for.

### 4.3 Practical Solution

Instead, following Thuburn et al. (2009), the coefficients can be uncoupled by defining

$$B_{v,v} = (\frac{R_{i,v}}{2} + C)n_{e,i} \tag{61}$$





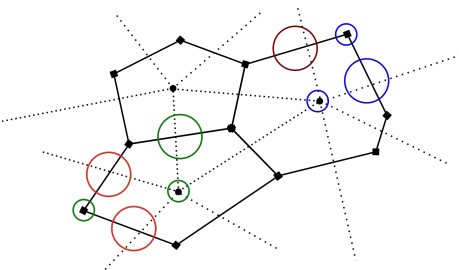

**Figure 5.** A diagram of the stencil $EVE(v,e,i) = EC(i) \cap EV(v) - e$. Consider the set $(v,e,i)$ denoted in green: then $EVE(v,e,i)$ are the two red edges. Now consider the set $(v,e,i)$ denoted in blue: then $EVE(v,e,i)$ is the brown edge.

$$B_{v,v'} = Cn_{e,i} \tag{62}$$

when $v \in VE(e)$ or $v' \in VE(e)$, where $C = -1/6$. On all meshes tested (including uniform square and uniform grid) there are enough degrees of freedom to do this, and the least-squares problem has a unique, exact solution. This has enabled the solution of the system for cubed-sphere meshes with up to 884736 grid cells and icosahedral-hexagonal meshes with up to 655363 grid cells in a few hours using an unoptimized, serial algorithm on a laptop computer. Furthermore, the uncoupled nature of the problem (one small independent least-squares problem per grid cell) would facilitate easy parallelism if needed for larger meshes (and again, the coefficients only need to be computed once).

### 4.3.1 PV Compatibility

The astute reader will note that nothing has been said yet about enforcing PV compatibility ($\mathbf{Q}_{q_v=c} = c\mathbf{W}$. It was originally believed that PV compatibility would have to added as additional equations in the matrix-vector system. However, it was found that enforcing potential enstrophy conservation (even using the cell split form) was sufficient to ensure that $\mathbf{Q}$ was PV compatible. This corresponds with the results of S04 (Salmon (2004)), who did not explicitly add PV compatibility, yet all of his schemes had this property. The reasons behind this result are not yet understood. If PV compatibility had to be added explicitly, it would simply mean that

$$\sum_{v \in VC(i)} \alpha_{e,e',v} = w_{e,e'} \tag{63}$$

for every edge pair $(e,e')$; which could be easily added to the independent system of equations solved in each grid cell.

## 5 Z Grid Scheme

Unlike the C grid scheme, the Z grid scheme starts with Nambu brackets rather than Poisson brackets. This greatly simplifies the derivation, since only the triply anti-symmetric nature of the brackets must be retained to ensure total energy and potential enstrophy conservation: there is no consideration of Casimirs. Start by defining a set of collocated discrete variables

$$\boldsymbol{x} = (h_i, \zeta_i, \mu_i) \tag{64}$$

which are pointwise values of $h$, $\zeta$ and $\mu$ at primal grid centers. More details about the grid, discrete operators and discrete variables can be found in Appendices A,D and F.

### 5.1 Functional Derivatives

The functional derivative of a general functional $\mathcal{F}$ with respect to discrete variable $x_i$ is then defined as

$$\frac{\delta \mathcal{F}}{\delta x_i} = \mathcal{F}_{x_i} = \frac{1}{A_i} \frac{\partial \mathcal{F}}{\partial x_i} \tag{65}$$

where $A_i$ is the area of primal grid cell $i$. The diagnostic variables $\Phi_i$, $\chi_i$, $\psi_i$ and $q_i$ are defined through the functional derivatives of the discrete Hamiltonian $\mathcal{H}$ and discrete Potential Enstrophy $\mathcal{Z}$ as:

$$\Phi_i \equiv \frac{\delta \mathcal{H}}{\delta h_i} \tag{66}$$


$$-\psi_i \equiv \frac{\delta \mathcal{H}}{\delta \zeta_i} \tag{67}$$

$$-\chi_i \equiv \frac{\delta \mathcal{H}}{\delta \mu_i} \tag{68}$$

$$q_i \equiv \frac{\delta \mathcal{Z}}{\delta \zeta_i} \tag{69}$$

At this point the discrete Hamiltonian $\mathcal{H}$ and discrete Potential Enstrophy $\mathcal{Z}$ are left unspecified.

### 5.2 Discrete Nambu Brackets

Following Salmon (2007), the general discretization starts from the Nambu brackets (33), (34) and (35) for the shallow water equations in vorticity-divergence form. As long as these brackets retain
their triply anti-symmetric structure when discretized, total energy and potential enstrophy will be automatically conserved for any definition of the total energy and potential enstrophy (with one




caveat explained below). In addition, the bracket structure ensures that this conservation is local as well as global. That is, the evolution of a conserved quantity can be written in flux-form for each grid cell, where cancellation of fluxes between adjacent cells leads to the global integral being invariant.

This is in contrast to a method that conserves the global integral, but cannot be written in flux-form for each grid cell. In what follows below, we will consider only the case where $\mathcal{Z}$ is the potential enstrophy, although this approach could be easily generalized to arbitrary Casimirs (see Salmon (2005) for an example of this on a uniform square grid).

### 5.2.1 Jacobian Brackets

Loosely following S07, the $\{\mathcal{F}, \mathcal{H}, \mathcal{Z}\}_{\zeta\zeta\zeta}$ bracket can be discretized as

$$\{\mathcal{F}, \mathcal{H}, \mathcal{Z}\}_{\zeta\zeta\zeta} = \frac{1}{3} \sum_{edges} \frac{1}{2} (D_1(\mathcal{Z}_\zeta)_v) J(\mathcal{F}_\zeta, \mathcal{H}_\zeta) + \text{cyc}(\mathcal{F}, \mathcal{H}, \mathcal{Z}) \tag{70}$$

Note that this bracket is triply anti-symmetric (due to the cyclic permutation), as required. The $\{\mathcal{F}, \mathcal{H}, \mathcal{Z}\}_{\mu\mu\zeta}$ bracket can be similarly discretized as

$$\{\mathcal{F}, \mathcal{H}, \mathcal{Z}\}_{\mu\mu\zeta} = \sum_{edges} \frac{1}{2} (D_1(\mathcal{Z}_\zeta)_v) J(\mathcal{F}_\mu, \mathcal{H}_\mu) \tag{71}$$

This bracket is only doubly anti-symmetric (in $\mathcal{H}$ and $\mathcal{F}$ due to the anti-symmetry of $J$), but it will conserve $\mathcal{Z}$ as well provided that $\frac{\delta \mathcal{Z}}{\delta \mu_i} = 0$ (since $J(A, B) = 0$ when either $A = 0$ or $B = 0$). These brackets are essentially those encountered when discretizing the Arakawa Jacobian, as detailed in Salmon (2005).

### 5.2.2 Mixed Bracket

The mixed bracket is trickier since it contains an apparent singularity ($\frac{1}{\overline{\nabla}q}$). On closer inspection, in the continuous case this singularity cancels out when combined with the functional derivative of the potential enstrophy. This is the caveat mentioned above- the discrete mixed bracket must be constructed such that the apparent singularity cancels out with the discrete functional derivative of the potential enstrophy. With this in mind, the general form of the discrete mixed bracket is chosen

as:

$$\{\mathcal{F}, \mathcal{H}, \mathcal{Z}\}_{\mu\zeta h} = \sum_{edges} \frac{\bar{D}_1(\mathcal{Z}_h)}{\bar{D}_1 q_i} \frac{le}{de} \left[ (\bar{D}_1 \mathcal{F}_\mu)(\bar{D}_1 \mathcal{H}_\zeta) - (\bar{D}_1 \mathcal{F}_\zeta)(\bar{D}_1 \mathcal{H}_\mu) \right] + \text{cyc}(\mathcal{F}, \mathcal{H}, \mathcal{Z}) \tag{72}$$

where, from before, $q_i \equiv \frac{\delta \mathcal{Z}}{\delta \zeta_i}$. This bracket is triply anti-symmetric (again due to the cyclic permutation), and the apparent singularity will cancel if **Z** is chosen with care.

### 5.2.3 Conservation

Since the $\{\mathcal{F}, \mathcal{H}, \mathcal{Z}\}_{\zeta\zeta\zeta}$ and $\{\mathcal{F}, \mathcal{H}, \mathcal{Z}\}_{\mu\zeta h}$ brackets are triply anti-symmetric, and the $\{\mathcal{F}, \mathcal{H}, \mathcal{Z}\}_{\zeta\mu\mu}$ bracket is doubly anti-symmetric, both total energy and potential enstrophy will be conserved for any choice of $\mathcal{H}$ and $\mathcal{Z}$; provided that the caveats mentioned above are obeyed. Those are:




1. $\frac{\delta \mathcal{Z}}{\delta \mu_i} = 0$ (ensures that the $\{\mathcal{F}, \mathcal{H}, \mathcal{Z}\}_{\zeta \mu \mu}$ bracket conserves potential enstrophy)

2. $\mathcal{Z}$ chosen such that the apparent singularity ($\frac{\bar{D}_1(\mathcal{Z}_h)}{\bar{D}_1 q_i}$ term + cyc($\mathcal{F}, \mathcal{H}, \mathcal{Z}$) terms) in the $\{\mathcal{F}, \mathcal{H}, \mathcal{Z}\}_{\mu \zeta h}$

   bracket cancels out

These are fairly minimal requirements, and many reasonable choices for $\mathcal{Z}$ satisfy them.

### 5.3 Discrete Hamiltonian and Helmholtz Decomposition

The Hamiltonian $\mathcal{H}$ can be split into three parts: $\mathcal{H}_{FD}$, $\mathcal{H}_J$ and $\mathcal{H}_{PE}$, where the first two are the kinetic energy due to flux-divergence terms and Jacobian terms, and the last is the potential energy. In the continuous system we have

$$\mathcal{H} = \mathcal{H}_{FD} + \mathcal{H}_J + \mathcal{H}_{PE} \tag{73}$$

where

$$\mathcal{H}_{FD} = \int_\Omega d\Omega \frac{1}{2h} [\boldsymbol{\nabla}\chi \cdot \boldsymbol{\nabla}\chi + \boldsymbol{\nabla}\psi \cdot \boldsymbol{\nabla}\psi] \tag{74}$$

$$\mathcal{H}_J = \int_\Omega d\Omega \frac{2J(\chi,\psi)}{2h} = \int_\Omega d\Omega \frac{J(\chi,\psi) - J(\psi,\chi)}{2h} \tag{75}$$

$$\mathcal{H}_{PE} = \int_\Omega d\Omega \frac{1}{2} gh(h + 2h_s) \tag{76}$$

These can be discretized as

$$\mathcal{H}_{FD} = \frac{1}{2} \sum_{edges} \frac{le}{de} \frac{(\bar{D}_1 \chi_i)^2}{h_e} + \frac{le}{de} \frac{(\bar{D}_1 \psi_i)^2}{h_e} \tag{77}$$

$$\mathcal{H}_{PE} = \frac{1}{2} \sum_{cells} A_i gh_i(h_i + b_i) \tag{78}$$

$$\mathcal{H}_J = \frac{1}{2} \sum_{edges} (D_1 \frac{1}{h_v}) J(\chi_i, \psi_i) \tag{79}$$

### 5.4 Helmholtz Decompositions and Bernoulli Function

By taking variations of $\mathcal{H}$ we obtain

$$\delta \mathcal{H}_{PE} = \sum_{cells} gA_i(h_i + b_i)\delta h_i \tag{80}$$





$$\delta\mathcal{H}_{FD} = \frac{1}{2}\sum_{edges}\frac{le}{de}\frac{(\bar{D}_1\chi_i)^2 + (\bar{D}_1\psi_i)^2}{h_e^2}\delta h_e + \sum_{edges}\frac{le}{de}\frac{(\bar{D}_1\chi_i)(\bar{D}_1\delta\chi_i)}{h_e} + \sum_{edges}\frac{le}{de}\frac{(\bar{D}_1\psi_i)(\bar{D}_1\delta\psi_i)}{h_e}$$

(81)

$$\delta\mathcal{H}_J = \frac{1}{2}\sum_{edges}D_1\frac{1}{h_v^2}\delta h_v J(\chi_i,\psi_i) + \frac{1}{2}\sum_{edges}D_1\frac{1}{h_v}\delta J(\chi_i,\psi_i)$$

(82)

After a lot of algebra, these can be grouped (half of each term involving $\delta h_i$ goes to $\Phi_i$ and half to $\mu_i/\zeta_i$) to obtain

$$\delta\mathcal{H} = -\chi_i\delta\mu_i + -\psi_i\delta\zeta_i + \Phi_i\delta h_i$$

(83)

where (using the definition of functional derivative)

$$\Phi_i = \frac{\delta\mathcal{H}}{\delta h_i} = \frac{1}{A_i}g(h_i+b_i) + \frac{1}{4}\frac{1}{A_i}\mathbf{K}\frac{le}{de}\frac{(\bar{D}_1\chi_i)^2 + (\bar{D}_1\psi_i)^2}{h_e^2} + \frac{C}{2}\frac{1}{A_i}\mathbf{K}D_1\frac{1}{h_v^2}J(\chi_i,\psi_i)$$

(84)

$$\mu_i = \frac{1}{A_i}D_2\frac{1}{h_e}\frac{le}{de}\bar{D}_1\chi_i - \frac{1}{2}\frac{1}{A_i}D_2(D_1\frac{1}{h_v})\psi_e$$

(85)

$$\zeta_i = \frac{1}{A_i}D_2\frac{1}{h_e}\frac{le}{de}\bar{D}_1\psi_i + \frac{1}{2}\frac{1}{A_i}D_2(D_1\frac{1}{h_v})\chi_e$$

(86)

The latter two equations (85 and 86) are the discrete version of the Helmholtz decomposition, and form a pair of non-singular elliptic equations. They can be combined into a single equation as

$$\mathbf{A}\begin{pmatrix}\chi_i\\\psi_i\end{pmatrix} = \begin{pmatrix}\mathbf{FD} & -\mathbf{JA}\\\mathbf{JA} & \mathbf{FD}\end{pmatrix}\begin{pmatrix}\chi_i\\\psi_i\end{pmatrix} = \begin{pmatrix}\mu_i\\\zeta_i\end{pmatrix}$$

(87)

where, for example, $\mathbf{FD}\chi_i = \frac{1}{A_i}D_2\frac{1}{h_e}\frac{le}{de}\bar{D}_1\chi_i$ and $\mathbf{JA}\psi_i = \frac{1}{2}\frac{1}{A_i}D_2(D_1\frac{1}{h_v})\psi_e$. Note that (without the $\frac{1}{A_i}$ factors) $\mathbf{FD}$ is symmetric and $\mathbf{JA}$ is anti-symmetric, which means that $\mathbf{A} = -\mathbf{A}^T$ (ie $\mathbf{A}$ itself is skew-symmetric). Also note that when $h_i = H$ is a constant (and therefore $h_e = H$), they reduce to

$$\mu_i = \frac{1}{H}\frac{1}{A_i}D_2\frac{le}{de}\bar{D}_1\chi_i = \frac{1}{H}\mathbf{L}\chi_i$$

(88)

$$\zeta_i = \frac{1}{H}\frac{1}{A_i}D_2\frac{le}{de}\bar{D}_1\psi_i = \frac{1}{H}\mathbf{L}\psi_i$$

(89)

where $\mathbf{L} = \frac{1}{A_i}D_2\frac{le}{de}\bar{D}_1$, which is the correct linearization behaviour.



### 5.5 Discrete Potential Enstrophy

A natural definition of the discrete potential enstrophy is

$$\mathcal{Z} = \frac{1}{2} \sum_{cells} A_i \frac{\eta_i^2}{h_i} \tag{90}$$

where $\eta_i = \zeta_i + f_i$. Taking variations of this yields

$$\frac{\delta \mathcal{Z}}{\delta \mu_i} = 0 \tag{91}$$

$$\frac{\delta \mathcal{Z}}{\delta h_i} = -\frac{1}{2} \frac{\eta_i^2}{h_i^2} \tag{92}$$

$$\frac{\delta \mathcal{Z}}{\delta \zeta_i} = \frac{\eta_i}{h_i} \tag{93}$$

Then the natural definition for $q_i = \frac{\eta_i}{h_i}$ works, and the above simplifies to

$$\mathcal{Z} = \frac{1}{2} \sum_{cells} A_i h_i q_i^2 \tag{94}$$

$$\frac{\delta \mathcal{Z}}{\delta h_i} = -\frac{1}{2} q_i^2 \tag{95}$$

$$\frac{\delta \mathcal{Z}}{\delta \zeta_i} = q_i \tag{96}$$

By plugging these back into the $\{\mathcal{F}, \mathcal{H}, \mathcal{Z}\}_{\mu \zeta h}$ bracket, it is seen that this choice of $\mathcal{Z}$ also ensures that the singularity cancels.

### 5.6 Independence between choices for $\mathcal{H}/\mathcal{Z}$ and Nambu Brackets

As noted before, the mimetic and conservation properties of the discrete scheme are completely in-
dependent of the choice of discrete Hamiltonian $\mathcal{H}$, provided the Hamiltonian is positive definite and produces invertible elliptic equations for the Helmholtz decomposition. If the resulting elliptic equations were singular, then the scheme would have a computational mode (as discussed in Salmon (2007)). Additionally, the discrete Helmholtz decomposition should also simplify to a pair of uncoupled Poisson problems when linearized. The mimetic and conservation properties are also
independent of the specific choice of $\mathcal{Z}$, provided that the singularity in the mixed bracket cancels and $\mathcal{Z}_\delta = 0$. The given choices of $\mathcal{H}$ and $\mathcal{Z}$ were selected to have these properties, and also correspond with those in S07 for the special cases of a uniform planar square grid and an orthogonal polygonal planar grid with a triangular dual.

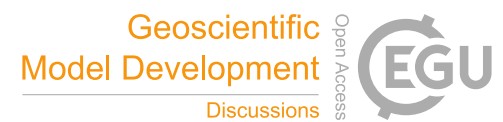



### 5.7 Discrete Evolution Equations

By setting $F = (h_i, \zeta_i, \mu_i)$ in turn, the following evolution equations are obtained:

$$\frac{\partial h_i}{\partial t} = -\mathbf{L}\chi_i \tag{97}$$

$$\frac{\partial \zeta_i}{\partial t} = \mathbf{J}_\zeta(q_i, \psi_i) - \mathbf{FD}(q_i, \chi_i) \tag{98}$$

$$\frac{\partial \mu_i}{\partial t} = -\mathbf{L}\Phi_i + \mathbf{J}_\delta(q_i, \chi_i) + \mathbf{FD}(q_i, \psi_i) \tag{99}$$

where $\mathbf{L}$ is the Laplacian, $\mathbf{FD}$ is the Flux-Divergence and $\mathbf{J}$ is the Jacobian. Note that these operators
on an icosahedral hexagonal-pentagonal grid are the same as those from Heikes and Randall (1995a).
The only difference is in the arguments ($q_i$ instead of $\eta_i$, and different definitions for $\chi_i$ and $\psi_i$.)

### 5.7.1 Laplacian and Flux-Div Operators

The Laplacian and Flux-Divergence operators (which come from the mixed bracket) can be written
as

$$\mathbf{L}\alpha_i = \frac{1}{A_i} D_2 \frac{le}{de} \bar{D}_1 \alpha_i \tag{100}$$

$$\mathbf{FD}(\alpha_i, \beta_i) = \frac{1}{A_i} D_2 \alpha_e \frac{le}{de} \bar{D}_1 \beta_i \tag{101}$$

where $\alpha_e = \sum_{i \in CE(e)} \frac{\alpha_i}{2}$.

### 5.7.2 Jacobian Operators

The Jacobian operators (which come from the Jacobian brackets) can be written as

$$\mathbf{J}_\delta(q_i, \chi_i) = -\frac{1}{A_i} D_2[(D_1 q_v)(\chi_e)] \tag{102}$$

$$\mathbf{J}_\zeta(q_i, \psi_i) = \frac{-1}{3}\frac{1}{A_i} D_2[(D_1 q_v)(\psi_e)] + \frac{1}{3}\frac{1}{A_i} D_2[(D_1 \psi_v)(q_e)] \tag{103}$$

Note that on a polygonal grid with a purely triangular dual (including the important case of an
icosahedral grid), $J_\delta = J_\zeta$.





### 5.8 Linearized Version

Under the assumption of linear variations around a state of rest ($h_i = H$, $\zeta_i = \mu_i = 0$, $q_i = \frac{f}{H}$) on a
f-plane, this scheme reduces to:

$$\frac{\partial h_i}{\partial t} = -\mathbf{L}\chi_i = -H\mu_i \tag{104}$$

$$\frac{\partial \zeta_i}{\partial t} = -\frac{f}{H}\mathbf{L}\chi_i = -f\mu_i \tag{105}$$

$$\frac{\partial \mu_i}{\partial t} = -g\mathbf{L}h_i + \frac{f}{H}\mathbf{L}\psi_i = -g\mathbf{L}h_i + f\zeta_i \tag{106}$$

where the Helmholtz equations given by (88) and (89) have been used to simplify the scheme (to the
point that it no longer requires solving any elliptic equations). In the case of a uniform square grid
(uniform hexagonal grid) this scheme is identical to the one studied in Randall (1994) (et. al (2002)),
and it shares the same excellent linear wave properties found for those schemes.

### 5.9 Relation to Salmon Schemes

For the cases of a uniform planar square grid and a general orthogonal planar polygonal grid with
triangular dual, the general discretization scheme presented above reduces to the schemes given in
S07. However, this discretization scheme is more general, and it also makes specific choices for the
total energy $\mathcal{H}$ and potential enstrophy $\mathcal{Z}$ when using a general polygonal grid.

### 5.10 Properties of Scheme

The discrete scheme as outlined above posses the following (among others) key properties:

1. Linear stability (Coriolis and pressure gradient forces conserve energy): Provided that $\mathbf{L} = \mathbf{L}^T$
   (which is satisfied for the $\mathbf{L}$ given above, and the majority of discrete Laplacians), the scheme
   will conserve energy in the linear case.

2. No spurious vorticity production: By construction, the pressure gradient term does not produce
   spurious vorticity since the curl is taken in the continuous system, prior to discretization.

3. Conservation: By construction, this scheme conserves mass, potential vorticity, total energy
   and potential enstrophy in both a local (flux-form) sense and global (integral) sense.

4. PV compatibility and consistency: By inspection, the mass-weighted potential vorticity equa-
   tion is a flux-form equation that ensures both local and global conservation of mass-weighted
   potential vorticity. In addition, an initially uniform potential vorticity field will remain uni-
   form. This rests on the fact that $\mathbf{J}_\zeta(q_i, \psi_i) = 0$ and $\mathbf{FD}(q_i, \chi_i) = c\mathbf{L}\chi_i$ when $q_i = c$ is con-
   stant.





5. Steady geostrophic modes: Since the same divergence $\mu_i$ appears in both the linearized vorticity and continuity equations, the scheme posses steady geostrophic modes.

6. Linear properties (dispersion relations, computational modes): As expected, the scheme possesses the same linear mode properties on uniform planar grids as those presented in Randall (1994) and et. al (2002); and it does not have any computational modes. More details of the linear mode properties of the scheme on both uniform planar and quasi-uniform spherical grids can be found in a forthcoming paper Eldred and Randall (20016b).

7. Accuracy: Unfortunately, as shown in Heikes et al. (2013), the Jacobian operator as given is inconsistent on general grids. Even more unfortunately, the fix proposed in that paper breaks key properties of the Jacobian necessary to retain total energy and potential enstrophy conservation. Surprisingly, as shown in Eldred and Randall (20016a), the inconsistency of the Jacobian operator does not appear to cause issues in the test cases that were run. More details on possible fixes to the accuracy issue are discussed in Eldred and Randall (20016a).

## 6 Conclusions

This paper presents an extension of AL81 to arbitrary non-orthogonal (spherical) polygonal grids in a manner that preserves almost all of the desirable properties of that scheme (including both total energy and potential enstrophy conservation) through a new $\mathbf{Q}$ operator. Unfortunately, on non-quadrilateral grids such as the icosahedral grid there will be extra branches of the dispersion relationship due to a mismatch in the number of degrees of freedom in the wind and mass fields inherent to the C grid approach. Switching from a C grid type staggering (to an A grid staggering, for example) is undesirable for many reasons, foremost among them being the natural association of physical variables with geometric entities in a staggered grid as suggested by exterior calculus and differential geometry (see Tonti (2014) and Blair Perot and Zusi (2014)). Fortunately, other than these extra mode branches on the icosahedral grid the proposed C grid scheme does not posses any additional computational modes. Furthermore, extensive testing has thus far been unable to show negative impacts from this extra mode branch, especially when running full-physics simulations with realistic topography and initial conditions (John Thuburn and Bill Skamarock, personal communication).

This work has also presented an extension of the total energy and potential enstrophy conserving Z grid scheme in S07 from planar grids to arbitrary orthogonal (spherical) polygonal grids, using the same toolkit of Nambu brackets and Hamiltonian methods. The restriction to orthogonal grids (geodesic grids are the only orthogonal quasi-uniform spherical grid the author is aware of) rather than more general non-orthogonal grids is a drawback. However, the major motivations for using a cubed-sphere grid are the ability to properly balance degrees of freedom when using a staggered C grid methods (and therefore avoid spurious branches of the dispersion relationship), a tensor-product grid structure for spectral or finite element type methods (which ensures a diagonal mass matrix for





spectral element methods and eases implementation of finite element methods) and higher-order
finite volume methods (enabling easy dimension splitting), and an underlying piecewise continuous
coordinate system for higher-order finite volume methods (allowing extended stencils). None of
these considerations apply to a Z grid method, so the restriction to icosahedral grids is not anticipated
to be a significant hurdle.

A detailed comparison of the two schemes, including an analysis of the accuracy of the operators
used and results from a variety of test cases can be found in second part of this series Eldred and
Randall (20016a). In addition, an analysis of the linear mode properties of these two schemes on
various quasi-uniform grids is undertaken in the third part of this paper series Eldred and Randall
(20016b).

## 7   Code Availability

The schemes described in this manuscript have been implemented in a Python/Fortran mixed lan-
guage code, and are freely available at https://bitbucket.org/chris_eldred/phd_thesis under a GNU
Lesser General Public License Version 3.

## Appendix A:  Discrete Grid

The schemes described above are designed to work on arbitrary (spherical) polygonal grids along
with an associated dual grid. In the case of the C grid scheme, the grid can be either orthogonal or
non-orthgonal, while the Z grid scheme is restricted to orthogonal grids. A description of the this
grid framework is given in what follows.

### A1   General Non-Orthogonal Polygonal Grid

Consider a (primal) conformal grid constructed of polygons (or spherical polygons). A dual grid is
constructed such that there is a unique one to one relationship between elements of the primal grid
and element of the dual grid: primal grid cells are associated with dual grid vertices, primal grid
edges are associated with dual grid edges and primal grid vertices are associated with dual grid cells.
This grid configuration covers the majority of grids that are used in current and upcoming atmo-
spheric dynamical cores, including cubed-sphere and icosahedral grids (both hexagonal-pentagonal
and triangular variants). Once the dual grid vertices have been placed, there are several important
geometric quantities that are needed in order to construct the discrete operators (shown graphically
in Figure 6). Specifically, we need the primal cell area $A_i$, the dual cell area $A_v$, the distance between
primal grid centers $le$, the distance between dual grid centers $de$ and the overlap areas $A_{iv}$ and $A_{ie}$.
On a planar grid, these are easily defined using the standard Euclidean metric and formulas. On a
spherical grid, distances must be calculated using geodesic arcs; and areas are calculated by subdi-





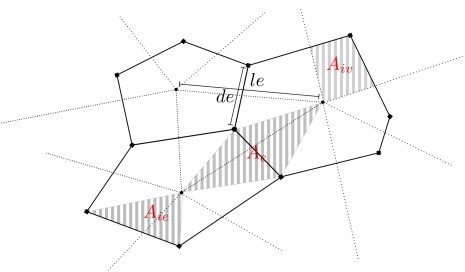

**Figure 6.** The geometric quantities on a planar grid. Primal grid edge lengths are denoted as $de$, dual grid edge lengths are denoted as $le$, the area associated with an edge by $A_e$, the overlap between primal grid cell $i$ and edge $e$ by $A_{ie}$ and the overlap between dual grid cell $v$ and edge $e$ by $A_{iv}$. Note that the same definitions can be used on a spherical grid, provided the appropriate measures are used (such as geodesic lengths for distances, and spherical polygonal areas for areas). See Weller (2013) for more details.

viding into spherical triangles as needed and then applying the relevant spherical area formulas. See the discussion in Weller (2013) for more details.

**Appendix B: Discrete Exterior Calculus Operators**

Following Thuburn and Cotter (2012), a set of discrete exterior derivative operators can be defined as:

$$D_1 = \sum_{v \in VE(e)} t_{e,v} \tag{B1}$$

$$\bar{D}_1 = \sum_{i \in CE(e)} -n_{e,i} \tag{B2}$$

$$D_2 = \sum_{e \in EC(i)} n_{e,i} \tag{B3}$$

$$\bar{D}_2 = \sum_{e \in EV(v)} t_{e,v} \tag{B4}$$

where $n_{e,i}$ is an indicator that is 1 when $e$ is oriented out of a primal grid cell and -1 when $e$ is oriented into a primal grid cell, and $t_{e,v}$ is an indicator that is 1 when $e$ is oriented into a dual grid cell and -1 when $e$ is oriented out of a dual grid cell. Note that by construction, these satisfy $D_2 D_1 = 0$, $\bar{D}_2 \bar{D}_1 = 0$, $D_2^T = -\bar{D}_1$ and $\bar{D}_2^T = D_1$ for arbitrary polygonal grids.





## Appendix C: Specific Choices for Various C Grid Operators

In order to close the C grid scheme presented in Section 3, specific choices must be made for $\mathbf{I}$, $\mathbf{J}$, $\mathbf{H}$, $\mathbf{R}$, $\phi$ and $\mathbf{W}$. The ones used here (and in Ringler et al. (2010) and Thuburn et al. (2013)) are:

$$\mathbf{I} = \frac{1}{A_i} \tag{C1}$$

$$\mathbf{H}_O = \frac{le}{de} \tag{C2}$$

$$\mathbf{H}_{NO} = \sum_{e' \neq e \in S(e)} H_{e,e'} \tag{C3}$$

$$\mathbf{J} = \frac{1}{A_v} \tag{C4}$$

$$\phi = \sum_{i \in CE(e)} \frac{A_{ie}}{A_e} \tag{C5}$$

$$\mathbf{R} = \sum_{i \in CV(v)} \frac{A_{iv}}{A_i} \tag{C6}$$

and

$$\mathbf{W} = \sum_{e' \in ECP(e)} W_{e,e'} \tag{C7}$$

where $\mathbf{H}_O$ is used on orthogonal grids such as the icosahedral grid, $\mathbf{H}_{NO}$ is used on non-orthogonal grids such as the cubed-sphere grid (the details of the construction of this operator, including the stencil $S(e)$ and the weights $H_{e,e'}$, can be found in Thuburn et al. (2013)) and the weights $W_{e,e'}$ are chosen such that $\mathbf{W} = -\mathbf{W}^T$ and $-\mathbf{R}D_2 = \bar{D}_2\mathbf{W}$ (the details for this operator can be found in Thuburn et al. (2009)). On an orthogonal grid, $\mathbf{I}$, $\mathbf{J}$, $\mathbf{H}$ correspond to the choice of a Voronoi hodge star from discrete exterior calculus.

## Appendix D: Specific Choices for Various Z Grid Operators

For the Z grid scheme, the following operators are needed:

$$K = \sum_{e \in EC(i)} \tag{D1}$$





$$J(A,B) = n_{e,2}A_2B_1 + n_{e,1}A_1B_2 \qquad \text{(D2)}$$

Note that $J(A,B)$ is anti-symmetric ($J(A,B) = -J(B,A)$) and satisfies $J(A,0) = J(B,0) = J(A,A) =$

0. In addition, two different interpolations (from cell centers to vertices and to edges, respectively)
are defined:

$$X_v = \sum_{i \in CV(v)} C X_i \qquad \text{(D3)}$$

$$X_e = \sum_{i \in CE(e)} \frac{1}{2} X_i \qquad \text{(D4)}$$

where $C$ is a constant given by $\frac{1}{n}$, where $n$ is the size of $CV(v)$ (equal to 4 for quadrilateral dual
grid cells and 3 for triangular dual grid cells).

**Appendix E: Discrete Variables (C Grid Scheme)**

Table 2 gives the discrete variables used in the C grid scheme, their type (which indicates the stagger-
ing on the grid), and their diagnostic equation (where applicable). For the type, the first designator

indicates the form type (primal or dual) and the second designator indicates the form degree (0,1
or 2). For example, $C_e$ is a dual 1-form. The only exceptions to this are the edge mass $m_e$, which
is used in constructing the dual mass flux $C_e$; and the edge PV $q_e$, which is used in constructing
**Q** for the variants that conserve only total energy or potential enstrophy. These quantities are not
really physical, but instead are just used computationally to construct other, physical quantities or

operators.

**Appendix F: Discrete Variables (Z Grid Scheme)**

Table 3 gives the discrete variables used in the Z grid scheme and their type (either prognostic or
diagnostic).

*Acknowledgements.* The authors would like to thank Pedro Peixoto for his helpful comments and suggestions

on an earlier draft of this manuscript. This work has been supported by the National Science Foundation Sci-
ence and Technology Center for Multi-Scale Modelling of Atmospheric Processes, managed by Colorado State
University under cooperative agreement No. ATM-0425247. Christopher Eldred was also supported by the De-
partment of Energy under grant DE-FG02-97ER25308 (as part of the DOE Computational Science Graduate
Fellowship administered by the Krell Institute).





**Table 2.** List of discrete variables and their diagnostic equations

| Variable | Type | Equation | Description |
|----------|------|----------|-------------|
| $m_i$ | p-2 | Prognostic | Mass |
| $u_e$ | d-1 | Prognostic | Wind |
| $b_i$ | p-2 | Constant | Topography |
| $f_v$ | d-2 | Constant | Coriolis Force |
| $C_e$ | d-1 | $C_e = m_e u_e$ | Dual Mass Flux |
| $F_e$ | p-1 | $F_e = \mathbf{H} C_e$ | Primal Mass Flux |
| $q_v$ | p-0 | $q_v = \eta_v / h_v$ | Potential Vorticity |
| $\zeta_v$ | d-2 | $\zeta_v = \bar{D}_2 u_e$ | Relative Vorticity |
| $\eta_v$ | d-2 | $\eta_v = \zeta_v + f_v$ | Absolute Vorticity |
| $\Phi_i$ | d-0 | $\Phi_i = \mathbf{I}(K_i + g m_i + g b_i)$ | Bernoulli Function |
| $m_v$ | d-2 | $m_v = \mathbf{R} m_i$ | Dual Mass |
| $\mu_i$ | p-2 | $\mu_i = D_2 \mathbf{H} u_e$ | Divergence |
| $K_i$ | p-2 | $K_i = \phi^T \frac{u_e^T \mathbf{H} u_e}{2}$ | Kinetic Energy |
| $\chi_i$ | d-0 | $D_2 \mathbf{H} \bar{D}_1 \chi_i = \mu_i$ | Velocity Potential |
| $\psi_v$ | p-0 | $-\bar{D}_2 \mathbf{H}^{-1} D_1 \psi_v = \zeta_v$ | Streamfunction |
| $m_e$ | e-0 | $m_e = \phi \mathbf{I} m_i$ | Edge Mass |
| $q_e$ | e-0 | Complicated | Edge PV |

**Table 3.** List of discrete variables and their diagnostic equations

| Variable | Type | Description |
|----------|------|-------------|
| $h_i$ | Prognostic | Fluid Height |
| $\zeta_i$ | Prognostic | Relative Vorticity |
| $\mu_i$ | Prognostic | Divergence |
| $\eta_i = \zeta_i + f_i$ | Diagnostic | Absolute Vorticity |
| $q_i = \eta_i / h_i$ | Diagnostic | Potential Vorticity |
| $\Phi_i = K_i + g h_i$ | Diagnostic | Bernoulli Function |
| $K_i$ | Diagnostic | Kinetic Energy |
| $\chi_i$ | Diagnostic | Velocity Potential |
| $\psi_i$ | Diagnostic | Streamfunction |

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
