# Peer review of "Total energy and potential enstrophy conserving schemes for the shallow water equations using Hamiltonian methods: Derivation and Properties (Part 1)"

_Geoscientific Model Development, 2016_

## Referee Comment (RC1) · Anonymous Referee #1 · 31 Oct 2016

Total energy and potential enstrophy conserving schemes for the shallow water equations using Hamiltonian methods: Derivation and properties (Part I) by C. Eldred and D Randall gmd-2016-238

General comments

The manuscript presents two new numerical methods for the rotating shallow water equations: one is an extension of the Arakawa and Lamb (1981) C-grid scheme to arbitrary non-orthogonal polygonal grids; the other is an extension of the Salmon (2007) scheme to arbitrary orthogonal polygonal grids. Their common feature is that they

are derived using Hamiltonian methods along with ideas from Discrete Exterior Calculus. This is an impressive piece of work and represents a significant step forward for such approaches. I also particularly enjoyed the Introduction, including Figure 1, which clearly and concisely summarizes the complex and often competing set of requirements that are considered desirable for numerical methods aimed (ultimately) at weather and climate modeling. The work will certainly be publishable after some revision

My main concern is that, especially towards the end, the presentation becomes less clear. Most of my comments are therefore aimed at improving the presentation so that readers can appreciate the substance of the paper.

—

Specific comments

My most important point is that throughout sections 3, 4, 5 (and Appendices B, C, D) the notation is awkward and potentially confusing. In the papers cited the notation $U$, for example (no subscript) is typically used to mean the vector comprising all the velocity degrees of freedom while the notation $U_e$ means one component of that vector, that is, the velocity at edge $e$. In the present manuscript $u_e$, for example, seems to have both meanings. This then makes many of the formulas difficult to understand, particularly for readers not intimately familiar with the cited previous work. It looks like it will be a rather tedious job to fix this up, but I think is it is essential for the clarity (and correctness) of the paper.

As the authors note, the principal novelty (and difficulty) of the proposed C-grid scheme is the specification of a suitable Q operator (section 4) given by some coefficients \alpha satisfying (60). (60) is eventually solved by a least squares method which (lines 441-2) 'has a unique, exact solution'. Could the authors please clarify whether (60) itself is solved exactly? If it is not then the scheme does not quite have the desired properties (though it may do so to a very good approximation). If, on the other hand,

(60) is solved exactly, despite being overdetermined, then this suggests that some solvability condition must be satisfied, as Thuburn et al 2009 found in constructing their W operator. (It might even be possible then to write down the solution fo the \alpha's without resorting to a numerical solution?) Either way, there should be something interesting to say.

Line 9, line 677 (perhaps elsewhere). Perhaps make it clear that here 'orthogonal' means that there is a dual grid whose edges are orthogonal to those of the primal grid.

Section 2.2. What is \Omega? Are some boundary conditions assumed in writing (7)?

P6. Define \nablaˆT; P7 define \nablaˆ\perp

Eq (26). There is the potential for some confusion because (26) is not quite the same expression as that below (2).

Line 204-241. Readers not familiar with differential forms and their discrete counterparts might be thrown by this new terminology. Perhaps explain briefly that 1-forms correspond to edge integrals and 2-forms to cell integrals; that should be enough for most readers to follow.

Eqs (37) (38). Explain the subscripts I and H.

Line 268. It is not clear until later that \phi is an interpolation operator.

Eq (50). Is there a factor 1/2 missing?

Section 3.4. From what we understand about the Hollingsworth instability, a Charney-Phillips vertical grid should reduced the instability compared to a Lorenz grid (rather than avoid it), but it could still be an issue at very high vertical resolution. Also, on a square grid one can rigorously derive a reformulation of the KE so as to rule out the Hollingsworth instability. On general grids this does not seem possible (e.g. Gassmann 2013), one can only minimize the non-cancellation that leads to the instabilty. In practice this seems to be sufficient at least for the published results, but we should not take

it for granted that the problem is solved.

Eq (84). Is K defined?

The definition of FD in (101) is not quite the same as that below (87), which could cause confusion.

Line 679. Surely any Voronoi tessellation / Delaunay triangulation gives an orthogonal primal-dual pair?

Appendices B and C. It would be helpful to have a brief (one or two sentence) interpretation of what each operator does.

—

Minor points, typos, etc

Line 37. typically -> typical

Line 40. Perhaps change 'realistic' to 'practical'?

Line 56. of a -> of

Line 65. posses -> possess

Line 111. \hat{k} is the unit vertical vector? I think standard notation would use bold font.

Line 132. posses -> possesses

Line 228. Use a consistent font for Z.

Line 252. I'm not sure Thuburn and Woollings (2005) is the correct reference here.

Line 253. Format for equation cross-references.

Line 281 and elsewhere. Reference to THESIS.

Line 289. pseudo-energy

Line 291. Coriolis parameter (f is not a force!)

Line 308. This is due.

Line 361. that the proposed

Line 378. which appendix?

Eq (72). Are le and de defined?

Eq (80). Is there a factor 1/2 missing?

Eq (81). Should the first term have a minus sign?

Line 591. Subscript \delta should be \mu ?

Line 655 and elsewhere. Eldred and Randall 20016a,b

Line 701. the this

---

## Referee Comment (RC2) · C. Cotter (Referee) · 4 Nov 2016

In this paper, the authors introduce some new aspects of C-grid and Z-grid schemes for the rotating shallow water equations on polygonal grids. The paper starts each section by collecting together quite a few bits of mathematical structure and previous results about C-grid and Z-grid schemes. These can be found elsewhere but it is nice to see them collected together in this context.

For C-grid schemes, they introduce a new methodology for obtaining schemes that simultaneously conserve energy and enstrophy on arbitrary orthogonal grids, a previously unsolved problem. The authors take the approach of writing out the (possibly over- or under-determined) system of equations that form the constraints, applying the Thuburn et al (2009) decoupling formula and solving the resulting system numerically. There is no proof of solvability for this system but the authors report that a unique solution is obtained in their numerical tests. For Z-grid schemes, the authors extend the Nambu bracket approach to arbitrary orthogonal grids, obtaining energy-enstrophy conserving schemes by construction.

As described in the introduction, this paper is Part I of a series of 3 papers with Part II containing numerical tests and Part III containing linear dispersion analysis. Part I concentrates on exposition of the methods and discussion of their properties. Given the focus of GMD on documenting and describing models and model software, I think that Part I requires:

(a) some evidence that the schemes are practically useful, i.e. that they do not do anything obviously weird. Having conservation properties is a good sign, but I have built plenty of schemes before that have good conservation properties but still lead to terrible numerics. If Part II promises to be a detailed comparison and analysis of results then Part I should at least have some provisional examples that show that things are working as expected.

(b) some evidence that the code provided is a correct implementation of the algorithms described.

(c) for GMD, I would expect some discussion as to how the numerical methods were implemented and expressed as code. This is not the same as code documentation, but should describe the main data structures used and how they form an efficient implementation.

It may actually be that a merge of Parts I and II makes sense, or some partial repetition between Parts I and II to achieve (a) and (b). Perhaps the solid rotation test plus something else where we can easily check that things are working, like the mountain

after the usual 12 days (or however long it is).

A few other remarks:

(1) It's bad form for a referee to ask for a reference to their own paper so feel free to ignore, but you might like to mention that the problem of simultaneously conserving enstrophy and energy on arbitrary grids was solved in the context of compatible finite element methods in McRae, Andrew TT, and Colin J. Cotter. "Energy‐and enstrophy‐conserving schemes for the shallow‐water equations, based on mimetic finite elements." Quarterly Journal of the Royal Meteorological Society 140.684 (2014): 2223-2234.

(2) The paper suddenly jumps in with 1-forms, 2-forms, Hodge stars etc without any warning to the reader! You should at least provide some references and a bit of a guide to what is going on, and maybe consider whether the language of differential forms is really necessary for this paper in terms of accessibility to a more general numerics audience.

(3) I would like to see some more description of how Equation (61) decouples the problem and how big the resulting uncoupled systems are. Why does it take so long to solve these systems? Why can't they be analysed to check if there is a unique solution?

(4) Please can you do a consistency test e.g. on the sphere for the Q operator? That is, take an analytic formula for u,h, interpolate to the grid and apply the Q operator, then analytically compute Q and interpolate to the grid, and compare errors in the L2 norm. I'd be especially interested in the cubed sphere case, where we observed lack of consistency for the Coriolis operator in our non-orthogonal scheme.

(5) If I'm thinking about this correctly, then the Q operator should imply a Coriolis reconstruction operator for the linear equations. Is this operator consistent in the limit as dx->0 on e.g. a cubed sphere mesh?
(6) There is no mention of timestepping anywhere. What do you do about timestepping in the code? How do time series of energy and enstrophy look?

(7) What is the relationship of the Z-grid scheme to Heikes et al (2013)? Is it a straight-forward extension of the same formulae to arbitrary grids or is another idea needed?

———————————————

---

## Author Comment (AC1) · 27 Dec 2016

The authors would like to start by thanking Anonymous Reviewer 1 for their helpful and thorough review, which has greatly improved the clarity, presentation and content of the manuscript. Responses to specific points raised in the review are given below.

My most important point is that throughout sections 3, 4, 5 (and Appendices B, C, D) the notation is awkward and potentially confusing. In the papers cited the notation $U$, for example (no subscript) is typically used to mean the vector comprising all the velocity degrees of freedom while the notation $U_e$ means one component of that vector, that is, the velocity at edge $e$. In the present manuscript $u_e$, for example, seems to have both meanings. This then makes many of the formulas difficult to understand, particularly for readers not intimately familiar with the cited previous work. It looks like it will be a rather tedious job to fix this up, but I think is it is essential for the clarity (and correctness) of the paper.

The notation has been modified to use uppercase to denote a vector of all degrees of freedom, while lowercase indicates a single component of the vector, and the subscript indicates the location of the degree of freedom. A hat is used to denote a quantity defined on the dual grid. For example, $\hat{U}$ denotes the vector of all velocity degrees of freedom, while $u_e$ indicates the velocity at edge $e$. The latter occur only within sums over relevant geometric entities, so the meaning of edge $e$ is unambiguous. The should help clarify the presentation in Sections 3,4,5 and Appendices B, C, D.

As the authors note, the principal novelty (and difficulty) of the proposed C-grid scheme is the specification of a suitable Q operator (section 4) given by some coefficients $\alpha$ satisfying (60). (60) is eventually solved by a least squares method which (lines 441-2) has a unique, exact solution. Could the authors please clarify whether (60) itself is solved exactly? If it is not then the scheme does not quite have the desired properties (though it may do so to a very good approximation). If, on the other hand, (60) is solved exactly, despite being overdetermined, then this suggests that some solvability condition must be satisfied, as Thuburn et al 2009 found in constructing their W operator. (It might even be possible then to write down the solution for the alphas without resorting to a numerical solution?) Either way, there should be something interesting to say.

Some additional discussion of the solution process for **Q** has been added to Section 4. Although the systems are overdetermined, an exact numerical solution is found (which was verified by checking that the defining relationships for energy and potential enstrophy conservation held with several sets of random vectors for $F_e$ and $q_v$). As you have mentioned, this implies the existence of a solvability condition; and it seems likely that determining the solvability condition would enable an explicit, analytic solution for the coefficients in terms of $R_{i,v}$ and $n_{e,i}$. Unfortunately, the authors were unable to determine the solvability condition for the case of general grids. This did not prevent, however, the successful use of a numerical solution to determine the coefficients.

Line 9, line 677 (perhaps elsewhere). Perhaps make it clear that here orthogonal means that there is a dual grid whose edges are orthogonal to those of the primal grid.

This has been clarified in the revised manuscript.

Section 2.2. What is $\Omega$? Are some boundary conditions assumed in writing (7)?

$\Omega$ represents the whole entire domain under consideration- either a doubly periodic plane

or the sphere. This implies that no boundary conditions are needed. This has been clarified in the revised manuscript.

P6. Define $\nabla^T$; P7 define $\nabla^\perp$

The presence of $\nabla^T$ was a typo. The skew gradient operator is defined as $\nabla^\perp = \vec{k} \times \nabla$ on the plane, where $\hat{k}$ is the unit vector in the vertical direction. Both it and the 2D curl $\nabla^\perp \cdot$ have a coordinate independent definition on more general manifolds. This has been clarified in the revised manuscript.

Eq (26). There is the potential for some confusion because (26) is not quite the same expression as that below (2).

A factor of $h_s$ was missing, thanks for catching this!

Line 204-241. Readers not familiar with differential forms and their discrete counter-parts might be thrown by this new terminology. Perhaps explain briefly that 1-forms correspond to edge integrals and 2-forms to cell integrals; that should be enough for most readers to follow.

To clarify the presentation, the references to differential forms and Hodge stars have been removed, and the relevant quantities have been redefined as integrals over the relevant geometric entities. A short section noting the relationship between the proposed C grid scheme and discrete exterior calculus has been added to provide more information for interested readers.

Eqs (37) (38). Explain the subscripts I and H.

These equations have been rewritten in a way that no longer requires the subscripts I and H, as a part of the change in notation.

Line 268. It is not clear until later that $\phi$ is an interpolation operator.

This has been clarified in the revised manuscript.

Eq (50). Is there a factor 1/2 missing?

No, when taking functional derivatives of the kinetic energy part of Hamiltonian the factor of 1/2 cancels a factor of 2 that comes from the square of velocity.

Section 3.4. From what we understand about the Hollingsworth instability, a Charney-Phillips vertical grid should reduced the instability compared to a Lorenz grid (rather than avoid it), but it could still be an issue at very high vertical resolution. Also, on a square grid one can rigorously derive a reformulation of the KE so as to rule out the Hollingsworth instability. On general grids this does not seem possible (e.g. Gassmann 2013), one can only minimize the non-cancellation that leads to the instabilty. In practice this seems to be sufficient at least for the published results, but we should not take it for granted that the problem is solved.

The discussion of the Hollingsworth instability has been revised in the updated manuscript. The principal points we are trying to make is that there appear to be several practical approaches to mitigating or avoiding the Hollingsworth instability for similar C grid schemes (reformulation of the kinetic energy expression, use of a Charney-Phillips staggering in the vertical, use of an isentropic or hybrid isentropic vertical coordinate); that these fixes do not affect the properties of the scheme such as energy conservation; and that therefore the possible presence of this instability should not prevent the use of the proposed scheme.

**K** is defined in the Appendix, this has been clarified in the revised manuscript.

The definition of FD in (101) is not quite the same as that below (87), which could cause confusion.

This has been changed in the revised manuscript.

Line 679. Surely any Voronoi tessellation / Delaunay triangulation gives an orthogonal primal-dual pair?

This is certainly true, and has been clarified in the revised manuscript. On the sphere, the only quadrilateral orthogonal grid the authors are aware of is the conformal cubed-sphere grid, which still suffers from resolution clustering at the panel corners. The more widely used gnomic cubed-sphere, which does not suffer from resolution clustering, is unfortunately non-orthogonal.

Appendices B and C. It would be helpful to have a brief (one or two sentence) interpretation of what each operator does.

These have been added in the revised manuscript.

Minor points, typos, etc

These have been fixed in the revised manuscript.

---

## Author Comment (AC2) · 27 Dec 2016

The authors would like to start by thanking Dr. Cotter for his helpful and thorough review, which has greatly improved the clarity, presentation and content of the manuscript. Responses to specific points raised in the review are given below.

In this paper, the authors introduce some new aspects of C-grid and Z-grid schemes for the rotating shallow water equations on polygonal grids. The paper starts each section by collecting together quite a few bits of mathematical structure and previous results about C-grid and Z-grid schemes. These can be found elsewhere but it is nice to see them collected together in this context.

For C-grid schemes, they introduce a new methodology for obtaining schemes that simultaneously conserve energy and enstrophy on arbitrary orthogonal grids, a previously unsolved problem. The authors take the approach of writing out the (possibly over- or under-determined) system of equations that form the constraints, applying the Thuburn et al (2009) decoupling formula and solving the resulting system numerically. There is no proof of solvability for this system but the authors report that a unique solution is obtained in their numerical tests. For Z-grid schemes, the authors extend the Nambu bracket approach to arbitrary orthogonal grids, obtaining energy-enstrophy conserving schemes by construction.

As described in the introduction, this paper is Part I of a series of 3 papers with Part II containing numerical tests and Part III containing linear dispersion analysis. Part I concentrates on exposition of the methods and discussion of their properties. Given the focus of GMD on documenting and describing models and model software, I think that Part I requires:

(a) some evidence that the schemes are practically useful, i.e. that they do not do anything obviously weird. Having conservation properties is a good sign, but I have built plenty of schemes before that have good conservation properties but still lead to terrible numerics. If Part II promises to be a detailed comparison and analysis of results then Part I should at least have some provisional examples that show that things are working as expected.

(b) some evidence that the code provided is a correct implementation of the algorithms described.

A short section (Section 6) has been added the paper with results from the Galewsky et. al ([3]) test case. The simulations look very similar to those obtained with other members of the TRiSK family, indicating that the new schemes are at least comparable. A detailed comparison will be performed in [2]. Evidence of the correctness of the implementation can be found in [1] (through an examination of the conservation properties), and this is mentioned in the revised manuscript.

(c) for GMD, I would expect some discussion as to how the numerical methods were implemented and expressed as code. This is not the same as code documentation, but should describe the main data structures used and how they form an efficient implementation. It may actually be that a merge of Parts I and II makes sense, or some partial repetition between Parts I and II to achieve (a) and (b). Perhaps the solid rotation test plus something else where we can easily check that things are working, like the mountain after the usual 12 days (or however long it is).

Also in Section 6, a brief discussion of the actual implementation has been added. It uses a combination of Python as a driver language with Fortran kernels for the numerics, and employs indirect addressing on a fully unstructured grid. Some thread-level parallelism in the numerics through OpenMP has also been added. This is not a particularly efficient implementation, although it served the purpose of demonstrating the properties of the proposed schemes.

A few other remarks: (1) Its bad form for a referee to ask for a reference to their own paper so feel free to ignore, but you might like to mention that the problem of simultaneously conserving enstrophy and energy on arbitrary grids was solved in the context of compatible finite element methods in McRae, Andrew TT, and Colin J. Cotter. "Energy and enstrophy conserving schemes for the shallow water equations, based on mimetic finite elements"

A reference to this paper has been added to the introduction, and some discussion of compatible finite elements also been included. The present work has been differentiated by emphasizing its use of finite-differences and explicit time stepping that does not require the inversion of a mass-matrix to compute the time tendency terms (although the vorticity-divergence scheme does require the solution of a Poisson problem at each time step to diagnose $\chi$ and $\psi$).

(2) The paper suddenly jumps in with 1-forms, 2-forms, Hodge stars etc without any warning to the reader! You should at least provide some references and a bit of a guide to what is going on, and maybe consider whether the language of differential forms is really necessary for this paper in terms of accessibility to a more general numerics audience.

To clarify the presentation, the references to differential forms and Hodge stars have been removed, and the relevant quantities have been redefined as integrals over the relevant geometric entities. A short section noting the relationship between the proposed C grid scheme and discrete exterior calculus has been added to provide more information for interested readers.

(3) I would like to see some more description of how Equation (61) decouples the problem and how big the resulting uncoupled systems are. Why does it take so long to solve these systems? Why cant they be analysed to check if there is a unique solution?

Some additional discussion of the solution process for $\mathbf{Q}$ has been added to Section 4. Once uncoupled, the resulting systems are quite small (24 coefficients for a square grid cell, 90 coefficients for a hexagonal grid cell), and they are very fast to solve. Although the systems are overdetermined, an exact numerical solution is found. This implies the existence of a solvability condition. As mentioned by reviewer 1, it seems likely that determining solvability condition would enable an explicit, analytic solution for the coefficients in terms of $R_{i,v}$ and $n_{e,i}$. Unfortunately, the authors were unable to determine the solvability condition for the case of general grids. This did not prevent, however, the successful use of a numerical solution to determine the coefficients.

(4) Please can you do a consistency test e.g. on the sphere for the Q operator? That is, take an analytic formula for u,h, interpolate to the grid and apply the Q operator, then analytically compute Q and interpolate to the grid, and compare errors in the L2 norm. Id be especially interested in the cubed sphere case, where we observed lack of consistency for

the Coriolis operator in our non-orthogonal scheme.

(5) If Im thinking about this correctly, then the Q operator should imply a Coriolis reconstruction operator for the linear equations. Is this operator consistent in the limit as dx→0 on e.g. a cubed sphere mesh?

The $\mathbf{Q}$ operator is designed to reduce (in the linear case) to the Coriolis operator $W$ from [4], since for a given $\mathbf{R}$ that is the unique operator that preserves steady geostrophic modes. The new $\mathbf{Q}$ therefore inherits all of the drawbacks of this operator, and in particular as you mentioned its inconsistency. This is a major issue with all TRiSK type schemes. This point has been clarified and further emphasized in the manuscript. A consistency check for $\mathbf{Q}$ is included in [2] and [1], confirming that it is indeed inconsistent on both icosahedral and cubed-sphere grids.

(6) There is no mention of timestepping anywhere. What do you do about timestepping in the code? How do time series of energy and enstrophy look?

This paper concerns itself only with spatial semi-discretization, and any stable time scheme can be employed with the spatial schemes presented here. For the tests performed in [2] and [1], 3rd order Adams-Bashford time stepping was used. The time series of energy and potential enstrophy look quite good, but of course the exact conservation implied by the spatial semi-discretization presented in this manuscript is lost.

(7) What is the relationship of the Z-grid scheme to Heikes et al (2013)? Is it a straight forward extension of the same formulae to arbitrary grids or is another idea needed?

The operators of the Z-grid scheme are the same as those of Heikes et. al (2013), simply with different arguments; and with a different Poisson problem used to diagnose $\chi$ and $\psi$ from $\zeta$ and $\mu$. These differences arise fundamentally from the use of a Hemholtz decomposition of the mass flux $h\vec{u}$ rather than the velocity $\vec{u}$. This point has been further emphasized and clarified in the revised manuscript.

**Bibliography**

[1] Christopher Eldred. *Linear and Nonlinear Properties of Numerical Methods for the Rotating Shallow Water Equations*. PhD thesis, Colorado State University, 2015.

[2] Christopher Eldred and David Randall. Total energy and potential enstrophy conserving schemes for the shallow water equations using hamiltonian methods: Test cases (part 2). In Preparation, 20016.

[3] Joseph Galewksy, Richard K. Scott, and Lorenzo M. Polvani. An initial-value problem for testing numerical models of the global shallow-water equations. *Tellus A*, 56(5):429–440, 2004.

[4] J. Thuburn, T.D. Ringler, W.C. Skamarock, and J.B. Klemp. Numerical representation of geostrophic modes on arbitrarily structured C-grids. *Journal of Computational Physics*, 228(22):8321–8335, December 2009.